

# Ventilation dynamics of the Oxygen Minimum Zone in the Arabian Sea

Henrike Schmidt[1,2], Rena Czeschel[1], Martin Visbeck[1,2]

[1]GEOMAR Helmholtz Centre for Ocean Research Kiel, Düsternbrooker Weg 20, 24105 Kiel, Germany

[2] Kiel University, Christian-Albrechts-Platz 4, 24118 Kiel, Germany

*Correspondence to*: H. Schmidt (hschmidt@geomar.de)

**Abstract.** Oxygen minimum zones (OMZs) in the open ocean occur below the surface in regions of weak ventilation and high biological productivity. Very low levels of dissolved oxygen affect marine life and alter biogeochemical cycles. One of

the most intense but least understood OMZs in the world is located in the Arabian Sea in a depth range between 300 to 1000 m. Within the last decades observations suggest a decreasing oxygen trend. Thus, an improved understanding of the crucial processes is necessary for a reliable assessment of the future development of the Arabian Sea OMZ.

This study uses a combination of observational data as well as reanalysis velocity fields from the ocean model Hycom (Hybrid Coordinate Ocean Model) to explore the ventilation dynamics of the Arabian Sea OMZ. Our results show that the

OMZ features a strong seasonal cycle with regional differences that is correlated with the monsoon system:

In the eastern basin, the OMZ is strongest during the winter monsoon with a core thickness of 1000 m depth and oxygen values of less than 5 μmol/kg. Ventilation during that phase is dominated by Persian Gulf water, that clockwise circles the perimeter of the basin and enters the OMZ from the north. During the summer monsoon ventilation from the southeast leads to higher oxygen values indicating a reverse flow along the Indian coast in the intermediate layer compared to the

southeastward surface currents.

The seasonal cycle in the western basin has the same seasonality as the one in the eastern basin with a core thickness of 900 m during the winter monsoon. The oxygen supply during the summer monsoon is weaker compared to the eastern basin and correlates with the ventilation of Persian Gulf (Red Sea) water during the summer monsoon (autumn inter-monsoon) phase.

As the interior exchange between the eastern and western basin is weak, the more pronounced OMZ in the eastern basin is

explained by prolonged ventilation time scales. For the eastern (western) basin Persian Gulf water needs 2-3 (1-2) years and Red Sea water 7-8 (3-4) years to ventilate the OMZ.



## 1 Introduction

Oxygen concentration at intermediate depth levels in the ocean is a result of the transport of oxygen from the surface mixed layer into the ocean interior (ventilation) and the local consumption of oxygen in the context of biological productivity and sinking organic matter (microbial respiration). In the south eastern parts of the tropical ocean poor ventilation south of the

subtropical gyre circulation (Luyten et al., 1983) and in the northern Indian ocean because the lack of ventilation from the north combined with high biological production in upwelling regions results in enhanced oxygen consumption by sinking organic matter and consequently very low levels of dissolved oxygen below the surface (Stramma et al., 2008; Gilly et al., 2013). These regions, so called oxygen minimum zones (OMZ) are characterized by low oxygen concentrations spanning a depth range of about 200-700 m depth (e.g. Karstensen et al. 2008).

It is well established that OMZs affect marine biogeochemical processes such as the global carbon and nutrient cycles (Bange et al., 2005; Naqvi et al., 2006). Conditions in near complete oxygen-depleted upwelling regions favour denitrification, which enhances production and release of climate-relevant trace gases to the atmosphere (Naqvi et al., 2010; Shenoy et al., 2012). Moreover, OMZs build a respiratory barrier in the subsurface layer impinging the ecosystem structure by limiting suitable habitats (Levin et al., 2009; Stramma et al., 2012; Resplandy et al., 2012).

Observations as well as global and regional models show a global trend towards decrease of oxygen and spatial expansion and intensification of OMZs during the last decades with noticeable regional variations (Stramma et al., 2008, 2010; Keeling et al., 2010; Diaz and Rosenberg, 2008). Declining oxygen is anticipated to intensify especially in coastal regions in response to global warming (Keeling et al., 2010; Schmidtko et al., 2017), which affects changes in ventilation, stratification, and solubility as well as eutrophication causing microbial respiration (Diaz and Rosenberg, 2008; Keeling et al., 2010, Breitburg

et al., 2018). Therefore, deoxygenation has become another major stressor affecting the marine and climate system besides warming and acidification and evolves into an important indicator for a changing oceanic environment.

Although there is no precise threshold where macro-organisms experience stress or die, or chemical cycles switch to alternative pathways the community has established four oxygen regimes and approximate thresholds. The boundary between oxic and hypoxic conditions is defined at 60 µmol/kg (Gray et al., 2002; Keeling et al., 2010). Regimes are termed

suboxic if oxygen concentration drops below 10 µmol/kg (Keeling et al., 2010) and nitrate involved respiration begins (Bange et al., 2005). Regions are called anoxic when dissolved oxygen drops below a few µmol/kg and sulphate reduction is the dominant respiratory process (Naqvi et al., 2010).

In this study we focus on the Arabian Sea OMZ (ASOMZ). It has the smallest horizontal extent of all oceanic OMZs, but is one of the most intense in the world tropical ocean based on the largest vertical extent of hypoxic water (Kamykowski and

Zentara, 1990) as well as on a significant core thickness with suboxic conditions of oxygen concentrations below 3 µmol/kg (Rao et al., 1994). The expansion of the ASOMZ is accompanied with declining sardine landings and an increase of fish kill incidents along the Omani coast (Piontkovski and Queste, 2016). Further expansion of the ASOMZ might have dramatic consequences on marine habitats and ecosystems (Keeling et al., 2010; Stramma et al., 2012). Hence food security and



livelihoods of one of the most populous regions on earth - about 25% of the world's population lives in the Indian Ocean rim countries - would be strongly affected (Breitburg et al., 2018). To understand ocean-climate interactions it is necessary to advance knowledge about the factors that impact the ventilation of the pronounced ASOMZ such as large-scale circulation.

While the circulation of the upper-ocean is fairly well known from drifter data (Shenoi et al., 1999) and satellite altimetry
(Beal et al., 2013) precise subsurface ventilation pathways of water masses entering the AS beneath the surface layer are less well understood in detail due to a lack of observational data (McCreary et al., 2013) and the complex interactions with the monsoon cycles. One unique difference of the Indian Ocean OMZs compared to the other ocean basins, that host OMZs is the fact that the upper layer of the Indian Ocean in general and the Arabian Sea (AS) in particular are strongly impacted by the Asian monsoon system resulting in a seasonally reversal of all boundary currents and associated ocean ventilation
patterns. Monsoonal wind forcing enabled by the land boundary in the north shifts from southwest winds during summer monsoon, causing strong upwelling off the coasts of Somalia and Oman, to northeast winds during winter monsoon driving downwelling circulation (Schott et al., 2001). The seasonal changes significantly influence biogeochemical cycles, biological activity and ecosystem response (Hood et al., 2009; Resplandy et al., 2012; Brewin et al., 2012).

Ventilation in the OMZ layers of the AS is facilitated by three major intermediate source water masses. Oxygenated Indian
Central Water (ICW) enters the AS at intermediate depth from the south (Fig. 1). High salinity Persian Gulf Water (PGW) enters the AS just beneath the thermocline in the north, spreading southward as well as along the perimeter of the basin (Prasad et al., 2001). Low salinity but denser Red Sea Water (RSW) enters the AS at intermediate depth and spreads across the basin (Beal et al., 2000; Shankar et al., 2005).

Several assumptions were made to explain the dynamical and biological processes associated with the shape of the ASOMZ.
So far it is known that, unlike in other tropical ocean basins, slow advection time is not responsible for the maintenance of the OMZ in the AS, where low-oxygen water has a residence time of 10 years (Olson et al., 1993). According to Sarma (2002) the residence time is even shorter with 6.5 years and the maintenance of the OMZ is caused by sluggish circulation combined with biological varying activities. Other studies explained the relatively high oxygen rates at the western boundary with the supply of oxygen-rich water transported by the western boundary current (Swallow, 1984; Sarma, 2002) and by
mixing of mesoscale eddies (Kim et al., 2001), although this is a region with high primary production at the surface and associated high consumption rates below. A process study of McCreary et al. (2013) stated the importance of the large-scale circulation for the shape of the ASOMZ as well as mesoscale features for variations of dynamical and biological processes.

Several studies have simulated the Indian Ocean circulation, whereby current model systems reveal large uncertainties and differences amongst them (McCreary et al., 2013). Typically, coarse resolution coupled biogeochemical ocean models
exhibit strong biases and tend to simulate lower oxygen concentrations in the Bay of Bengal than in the AS (e.g. Oschlies et al., 2008) contradicting the existing observations. Lachkar et al. (2016), however, suggests that the model performance improves with increasing model resolution. The latter findings support suggestions by Resplandy et al. (2012) and McCreary et al. (2013) that horizontal eddy mixing strongly impacts the oxygen dynamics in the AS. Studies on the equatorial Pacific from global coupled biogeochemical circulation models (Dietze and Löptien, 2013) point out that poor model performance is





related to a deficient representation of ventilation pathways rather than being associated with a deficient representation of biogeochemical processes (i.e. respiration). This confirms the need for a better understanding of the intermediate circulation in the AS including the pathways of RSW and PGW to understand the associated variability of the ASOMZ and related climate-biogeochemical interactions.

It remains an open question how the interplay between physical and biogeochemical processes influences the Indian Ocean oxygen dynamics. Specifically, there are two issues for the ASOMZ:

1) Why does the ASOMZ occur further east relative to the upwelling area with associated high productivity? A good explanation of this eastward shift could not be given so far (Acharya and Panigrahi, 2016).

2) Why is the ASOMZ maintained throughout the year with only a very weak seasonal cycle compared to the dramatic

changes of physical forcing and biogeochemical conditions associated with the seasonal reversing monsoon winds?

The present study focuses on advective pathways relevant for the ventilation dynamics of the ASOMZ. The circulation on the isopycnal layer of $\sigma = 27$ kg/m$^3$ was chosen because it covers the upper core of the ASOMZ. A backward-trajectory analysis was applied to examine the source regions of the seasonally changing ventilation pathways of the major water masses in the ASOMZ.

The following section explains the data sets and methods used for this study. Section 3 presents main ventilation pathways for the eastern and western basin of the AS, as well as their time scales and seasonality that are relevant for the variability of the ASOMZ and associated uncertainties. This is followed by the discussion and summary in section 4.

## 2 Data and Methods

The study uses the global dissolved oxygen climatology of the World Ocean Atlas 2013 (WOA13) as observational data.

The monthly mean data cover a period from 1955-2012 and are available with a spatial resolution of 1° x 1° interpolated on 102 depth levels (Garcia et al., 2013).

The core of the OMZ is defined by water masses having a dissolved oxygen concentration of less than 10 $\mu$mol/kg (Fig. 2, 3). The seasonal oxygen dynamics within this core region are the basis for the backward trajectory experiment setup.

Trajectory calculations are based on reanalysis velocity data from the dynamic ocean model Hycom (Hybrid Coordinate

Ocean Model) (Bleck, 2002) provided by the Center for Ocean-Atmospheric Prediction Studies (COAPS). The model has a spatial resolution of 1/12° in longitude and latitude with 40 depth levels between 0 and 5000 m, with decreasing resolution towards greater depth from 2-1000 m. It has a realistic bathymetry based on the General Bathymetric Chart of the Oceans (GEBCO) and uses isopycnal coordinates in the open, stratified ocean, changes to terrain-following coordinates in shallower and coastal regions and uses z-level coordinates in the mixed layer. The surface forcing from the National Center for

Environmental Prediction (NCEP) Climate Forecast System Reanalysis (CFSR) is used in the time between 1995-2012. Furthermore, these Hycom is run in data assimilation mode using gridded 'observations' from the Navy Coupled Ocean Data Assimilation (NCODA) system (Cummings, 2005; Cummings and Smedstad, 2013).





The variable vertical coordinates are beneficial for the model to better reproduce the circulation near the out-/overflow regions compared to typical z-level models, which would generally have problems to resolve the shallower coastal regions properly (see also Bleck and Boudra, 1981; Bleck and Benjamin, 1993; Bleck, 2002). The latter is important for the analysis of the supply of Persian Gulf Water (PGW) and Red Sea Water (RSW) through the Gulf of Oman and the Gulf of Aden respectively.

The model velocity output available to use were daily snapshots for the time period from January 2000 to December 2012. The velocity field during winter and summer monsoon is shown using the mean seasonal velocity for the months November to February and June to September, respectively, averaged for the years 2000 to 2012. The data are spatially filtered using a 0.6° x 0.6° window and presented on a grid with the same resolution (Fig. 5).

## 2.1 Trajectory computation

The two dimensional Lagrangian trajectories were computed offline from Hycom reanalysis velocity fields following Fischer (2007). First the velocity fields were vertically interpolated onto the isopycnal of 27 kg/m$^3$ using linear interpolation. This two dimensional approach is very time efficient but ignores the effects of upwelling or diapycnal mixing. In the AS, the supply of oxygen on isopycnal surfaces of 27 kg/m$^3$ was first suggested by Banse et al. (2014) for the layer between 300-500 m depth. The choice of this isopycnal had two main reasons.

1.      The core densities of PGW ($\sigma$ = 26.4 kg/m$^3$) and RSW ($\sigma$ = 27.4 kg/m$^3$), which appear to be the main source water masses ventilating the ASOMZ, are close to the isopycnal density of $\sigma$ = 27 kg/m$^3$.

2.      These isopycnal lies in the upper core of the ASOMZ (Fig. 3). Furthermore, this is the density layer with large interannual changes in oxygen concentration (Fig. 4a) and values of less than 10 $\mu$mol/kg nearly everywhere throughout the year (Fig. 4a).

Trajectories are calculated with an Euler forward-in-time integration of daily velocity fields using a time step of 1/20 day. With the chronologically updated velocity field, this method gives the downstream trajectories of the particles (forward trajectories). The velocity field is also updated in reversal time and particles are moved into the opposite direction to compute the upstream path of the fluid parcels (backward trajectories), therefore yielding the source regions. Further analyses are done using particle positions with a time step of 4 days.

In addition to the velocity field a random walk of particles is applied to consider subscale diffusion of 20 m$^2$/s. Near the coast lines a special case of random walk in offshore direction is used to prevent trajectories entering land. The choice of magnitude of random walk is connected to the spatial and temporal grid resolution. A series of sensitivity runs were performed and for example three runs performed with different subscale diffusion coefficients of 10 (run 28), 20 (run 27) and 25 (run 29) m$^2$/s (Tab. 1) do not reveal significant different results (not shown here). Nevertheless, there are some grid boxes along the coastline and especially at islands, where the particles get trapped. These spuriously high probabilities were not

considered for further analyses. On the other hand, the velocity fields of Hycom are not divergence/convergence free, also in coastal regions and near islands (e.g. the Maledives, Socotra). It is important to consider these regions for interpretation.

The launching positions of the backward trajectories in the eastern (ER) and western (WR) part of the core of the ASOMZ as well as for the forward trajectory launching points in the marginal seas, Red Sea (RS) and Persian Gulf (PG), are shown in

Figure 1. An overview over the performed runs is given in Table 1.

## 2.2 Trajectory visualization

To analyse the Lagrangian data, the AS is divided into a grid of 1° x 1° resolution. For each time step the number of particles residing in a certain grid box can be counted leading to a map that shows the particle concentration over the analysis time in

a certain grid box (Gary et al., 2011). For a better comparison, the particle counts are normalised by the total number of particles divided by the number of boxes that can be occupied by the particles.

With a subsample of the trajectories that reach the source regions, these maps can highlight the most likely taken Lagrangian particle pathways (Gary et al., 2011). On the other hand, it is possible to analyse the spreading of the particles by looking at single time steps. The transit time is analysed along cross sections on the ventilation pathways. Therefore, no particle is

counted twice and only the first crossing time of each particle at each section is detected. Along these sections a seasonal cycle of particle crossings can be determined.

## 2.3 Trajectory validation and statistics

To test the reliability of the calculated Lagrangian trajectories (Tab. 1, runs 1 and 6) multiple model runs with identical setup

were performed (runs 1-5). Starting the calculation in a different year (runs 17-18 and 30-35) or using a daily climatological velocity field (runs 7-8), which contains the mean velocity of the 13 years at each grid point and day will give an insight on the quality of the results.

Seasonal differences can be predicted by starting the calculations with a lack of 3 month (January, April, July and October) and running them for 1 year only (runs 9-16 and 19-26).

For reasons of computational cost savings, the quality of the results by using a smaller number of particles was validated. A comparison between 50000 and 10000 fluid parcels confirmed similarly good results (not shown here), so runs 27 to 35 were performed with fewer particles to save cost.

The ventilation time and seasonality was analysed with three runs for each release (runs 30-35) passing cross sections along the main particle pathways (see also section 2.2).





## 3 Results

### 3.1 Seasonal oxygen dynamics and circulation at intermediate depth

The northern Indian Ocean is a region of strong monsoonal forcing and it is known that seasonal changes have a profound impact on the ASOMZ (Resplandy et al., 2012). Thus, first we give a short overview of the seasonal variability of the
suboxic oxygen distribution and the circulation at intermediate depth in the AS.

The OMZ in the northern AS has a strong seasonal variability with regional differences, especially in the upper core (350 – 550 m depth, Fig. 4). The annual mean of oxygen from observational climatologies shows that the layer containing oxygen of less than 10 µmol/kg is deepest in the eastern basin (Fig. 2a). The maximum thickness arises during the winter monsoon with a depth of 1000 m (Fig. 4b) and nearly total oxygen depletion in the core (Fig. 4a). Oxygen concentration increases
within spring inter-monsoon and the suboxic layer in the eastern AS nearly vanishes in May (Fig. 4b).

A similar seasonal cycle is prominent in the western AS with a maximum thickness of the suboxic layer of 900 m (Fig. 4b) but however, a weaker ventilation during the summer monsoon compared to the east. The layer containing oxygen of less than 10 µmol/kg remains thicker than 500 m. Based on an area of 2°x 2° in total around the release location the spatial standard deviations were calculated (Fig. 4). They show that the seasonal cycle of the OMZ represents a large area and not
only the release location. This holds especially for the eastern basin.

Lagrangian trajectories were calculated on basis of the daily Hycom reanalysis velocities on the isopycnal surface of 27 kg/m$^3$. Also at intermediate depth in the AS several boundary currents are seasonally changing directions such as the Somali Current along the western boundary. Mean seasonal velocity for the period of 13 years shows the reversing of the Somali Current from a southwestward boundary current during winter monsoon (Fig. 5a) to a stronger northeastward boundary
current during summer monsoon (Fig. 5b). The annual mean of the Somali Current reveals a northeastward boundary current. Generally, velocities show the strongest variability along the boundaries, especially in the western basin increasing towards the equator (Fig. 5c) and in the marginal seas. Along the eastern boundary of the AS the flow at intermediate depth is also changing its direction between the different monsoon phases from a distinct southeastward directed flow along the west coast of India during the northeast monsoon (Fig. 5a) to a northwestward directed more variable flow during the
southwest monsoon (Fig. 5b).

### 3.2 Particle origins and main pathways

The strong contrast in extension and seasonal cycle of the ASOMZ for the eastern and western basin encourages to analyse the ventilation of each half of the basin individually. Therefore, we define two release locations in the eastern (ER) and
western (WR) part of the core of the ASOMZ (Fig. 1). Trajectories for 50000 particles were calculated backward for each of the release locations over the period of 13 years (runs 1 and 6). In the following we present the main pathways at intermediate depth within the AS for the eastern and western basin of the AS.



Particles that ventilate the eastern part of the ASOMZ show highest probability occurring east and northeast of the launching area along the North Indian and Pakistani coastline (Fig. 6b). There is also a northward advection of particles along the coast of India from the southeastern part of the AS. Both ventilation paths along the eastern boundary of the AS are mirrored in the seasonal mean of the intermediate circulation for the winter and summer monsoon (Fig. 5a, b). Particles travelling from the
western AS into the eastern OMZ predominantly emerge in the Gulf of Oman and off the Omani coastline (Fig. 6b). The patterns of particle distribution after only 4 years (see supplement Fig. S1b) reveal similar results of particle appearance as the histogram over the whole time series (Fig. 6b). Simulations of another 4 years backward show a wider and more equally spread distribution of particle origins that ventilate the eastern part of the ASOMZ (Fig. S1d).

For the western part of the ASOMZ the ventilating particles have the highest probability to occur north of the launching
position (Fig. 6a) but more equally spread in all directions around the release point than it is the case in the eastern basin, which is also reflected in the high variability of the velocity field in the northwest corner of the AS (Fig. 5c). Along the western boundary southward particle advection is distinct from the Gulf of Oman and the PG and northward particle advection from the RS into the western ASOMZ (Fig. 6a). The snapshot of the particle distribution after 4 years shows the origin of ventilating particles also from the eastern part of the ASOMZ (Fig. S1a). Similar to the ER, the particle origins after
8 years of simulation spread wider and more equally over the AS (Fig. S1c). After 12 years of simulation the particle origins for ventilation reveal no fundamental differences between the WR and ER (Fig. S1e-f).

For two of the three major intermediate source water masses that ventilate the ASOMZ - the RSW and the PGW - the source area can be localized clearly so that it is possible to extract the particles that origin from the RS and the PG. The most prominent pathway of fluid particles from the PG and the RS circles the basin clockwise along the western, northern and
northeastern boundary into the eastern ASOMZ (Fig. 7b, d). This is valid for RSW as well as PGW, which enters the AS in the north through the Gulf of Oman (Fig. 7b). The more direct interior pathway is less important, especially for the RSW (Fig. 7d). RSW spreads mostly notheastward along the coast of Yemen/Oman, where it enters the western part of the ASOMZ (Fig. 7c). Instead of flowing further eastward at along 20°N, most particles spread further north along the coastline of Pakistan and India to reach the eastern basin. The pathway of PGW into the western part of the ASOMZ also tends to flow
along the coastline off Oman (Fig. 7a) but is directed southward.

Water entering from the south at intermediate depth (ICW) shows a direct exchange from the western to the eastern basin in the region of the OMZ (Fig. S2). It is also remarkable that the particles enter the AS more from the southeast and tend not to follow the western boundary current.

### 3.3 Ventilation time

After analysing the main pathways, in the following we focus on the ventilation time of the virtual particles helping to further understand the circulation at intermediate depth. Therefore, the point-to-point transit time of the particles across





selected sections along their distinct pathways is analysed (Fig. 8). The first, respectively highest peak is considered as the transit time with the strongest particle flow.

The western part of the ASOMZ is ventilated preferably from particles coming from the northern basin. Within the first six months 17% of the released particles travel the pathway along the western boundary between the 21°N section and the WR

revealing the shortest transit time for this region (Fig. 8a).

The number of particles travelling northward is weaker: transit time peaks after one year when about 2% of the released particles travelled from the section at 17°N to the western release point (Fig. 8a).

In contrast, in the eastern basin the numbers of particles ventilating the eastern part of the OMZ from the northern and the southern section are about the same size with rates of 7% (17°N) and 9% (21°N) of the released particles (Fig. 8b). Particles

released in the eastern part of the ASOMZ reveal a duration of less than one year for the distance from the northern section at 21°N to the ER (Fig. 8b). After one year, the number of particles crossing that latitude weakens. Particles travelling from the southern section (17°N) reveal a shorter transit time of <0.5 year, which is probably related to the monsoon circulation.

2% of the released particles are travelling around the perimeter of the basin (Fig. 8b), which is comparable to the amount of particles taking the interior pathway between ER and WR (Fig. 8c, d). The transit time of particles travelling the direct route

in the interior basin is shorter for the eastward direction (8 months, Fig. 8d) than for the westward pathway (1 year, Fig. 8c). Particles travelling around the perimeter of the basin of the AS reveal a transit time up to 2 years from the section at 64°E (Fig. 8b).

Transit times from the source region of the PG as well as from the RS are faster to the western basin (1 year and 3.5 years, respectively; Fig. 8c) than to the eastern basin (4 years and 5.6 years, respectively; Fig. 8d), due to their shorter distance. The

number of particles travelling from the marginal seas to the WR and ER, respectively is less than 1%.

The Arabian Sea is also ventilated from the south across 10°N (Fig. 8e, f). For both release points, WR and ER, the ventilation is stronger across the eastern half of the basin (Fig. 8e) and takes 4.3 years.

Due to a broad distribution, the number of particles that cross the sections decreases with the distance of the particles from their release point. The spreading is faster from the PG to the WR (Fig. 8c), and more particles stream into the western AS

compared to the eastern AS (Fig. 8d). Also the ventilation of the western basin from the PG exceeds the ventilation of the RS (Fig. 8c).

### 3.4 Seasonal Variability

The seasonal variability of certain pathways is shown by the monthly average of the percentage of particles (Fig. 9). One of

the strongest seasonal variability is revealed by the southward pathways between the sections at 21°N and the WR as well as the ER, however, with a more distinct amplitude in the western basin (Fig. 9a, b). The WR is mainly ventilated during the inter-monsoon phase in spring by southward travelling particles from the northern part of the AS (Fig. 9a). This maximum coincides with the oxygen maximum in the western basin (Fig. 4a).





In contrast, in the eastern basin particles move preferably southward between the section at 21°N and the ER during the winter monsoon (Fig. 9b), which is also reflected in the southward eastern boundary current at intermediate depth (Fig. 5a). The eastward movement along the northern boundary across 64°E is also strongest during the winter monsoon (Fig. 9c). Therefore, the minimum of oxygen concentration in the eastern basin in winter (Fig. 4a) might be explained by the longer

ventilation time while looping around the northern part of the basin and crossing regions with high primary production and resulting high consumption rates.

While the eastward transport of particles along the northern boundary is weakest in spring inter-monsoon season (Fig. 9b), at the same time a higher direct interior transport from the western to the eastern half of the ASOMZ can be observed (Fig. 9d). The northward transport into the AS across 10°N mainly takes place at the eastern part of the basin showing a seasonal cycle

which is also highest during the spring inter-monsoon season (Fig. 9e, f). Hence, the maximum oxygen concentration at intermediate depth in spring (Fig. 4a) might be associated with the northward and eastward ventilation of the eastern basin, which peaks during the inter-monsoon phase in this region (Fig. 9b, d, f). The northward transport into the AS is weak at the western side of the basin and reveals a small cycle depending on the reversing monsoon (Fig. 9e, f).

The particle spreading based on climatological velocity (runs 9-16, Fig. 10) also reveals seasonal variability. Particles that

travel backwards from the ER for 1 year starting in March (Fig. 10b) and September (Fig. 10d) show similar probability maps of their particle position and do not reveal any significant difference compared to other season releases (not shown here). Nevertheless, spreading distances calculated over one season indicate higher values between April and September ($5.2\text{-}6.3 \times 10^6$ km) than between October and March ($2.9\text{-}4.9 \times 10^6$ km). This corresponds with a strong ventilation along the eastern boundary during the southwest monsoon showing the highest ventilation rates from the south across 17°N (Fig. 9b)

in June reflecting the northward current at the eastern boundary during the summer monsoon (Fig. 5b). But there are smaller peaks in October and March as well, so that the only period with weak ventilation into the eastern basin appears during the late summer monsoon and the winter monsoon (Fig. 9b).

For the western basin (WR), travel distances are also higher during the summer monsoon and here the patterns differ along the coastline. Fluid parcels that ventilate the OMZ in spring originate from the southwest (Fig. 10a), whereas the ventilation

in autumn, shortly before the onset of the winter monsoon, is more pronounced from the northeast and northwest (Fig. 10c). This indicates an influence of the turning western boundary current on the ventilation of the western ASOMZ.

However, the seasonal pattern reveals that the southward transport from the PG is strongest during spring and the transport out of the RS peaks during the autumn inter-monsoon season (Fig. 9c). Nevertheless, the actual northward transport of these water mass does not show a distinct seasonal cycle and so does the interior transport from the eastern OMZ into the western

OMZ (Fig. 9c).

The patterns of particles released in the source regions during the inter-monsoon season show, that particles spread into different directions during summer and winter monsoon (Fig. 11). Particles from the PG spread furthest during summer monsoon and shortly after, with $10.2 \times 10^6$ km spreading distance between July and September and $9 \times 10^6$ km spreading distance between October and December. Furthermore, the spreading direction changes with monsoon seasons. Between





June and October pathways go further into the mid AS (Fig. 11c) and along the northern coast, whereas during the winter monsoon particles stay closer to the coast and travel southward (Fig. 11a).

Particles from the RS show nearly the same behaviour as particles from the PG as pathways in both experiments seem to be influenced by the wind stress and/or surface currents (Fig. 11b, d), since the particles are pushed out of the Gulf of Aden in summer and autumn, then turning southward with the reversing winds. The maximum spreading distance lies again between July and September with $9.1 \times 10^6$ km.

Furthermore, the time series of the particle distances are not exclusively increasing in forward as well as in backward experiments (not shown here), which indicates recirculation of the trajectories either by the turning main current systems with monsoon shift or by small scale circulation like eddies.

### 3.5 Trajectory error estimation

One source of error arises from the calculation technique itself by adding a subscale diffusion and the random walk at the coastlines. To predict the discrepancy of that error, runs 1-5 were performed with identical setup. The percentage of trajectories reaching the PG/RS and southern IO and their mean travel times (Tab. 1) have standard deviations of 0.13, 0.01, 0.2/0.21 for PG particle percentage, RS particle percentage and ICW particle percentage (east/west), respectively, which is not distinguishable from the histogram maps.

As earlier discussed in Section 2.1, a different value for the diffusivity coefficient does not change the results significantly. The number of particles used for the first 26 runs (Tab. 1) is higher than it has to be to yield equally good results (Sect. 2.3). Further runs were performed with climatological velocity values over the same time interval of 13 years (runs 7 and 8). The particle probability maps for these simulations do not differ from the ones shown in Figure 6. Nonetheless the particle percentages and mean times for the source regions (Tab. 1) are different. The maximal difference is noticeable with a deviation of 30 % for the particles entering the PG from the western part of the ASOMZ (runs 8 and 18). Other runs show discrepancies between 4-6 % in particle percentages.

A year-to-year time series analysis of the velocities shows strong damping for the climatology having peak velocities of 0.2 compared to about 0.8 m/s (not shown here).

To test the reliability over time, runs with the length of 8 years were performed, each starting with an offset of 2 years (runs 30-35, Tab. 1). Again, the particle probability maps show similar results among each other (not shown here) as well as in comparison with the longer runs shown in Figure 6. Concerning the standard deviations, values lie between 0.3 for the RS-ER and 8.4 for the eastern ICW- ER particle percentage. These huge differences let suspect a dependency on small scale, short time variability in the experiment. Therefore, the seasonality analysis was performed with the 6 runs over 8 years to get a more confident result and better travel times, smoothing out possible burst of years with strong ventilating currents.





Due to the small number of runs, it is not possible to give an estimation error, but the range in which order the values spread is given as discussed above.

## 4 Discussion and Conclusions

Focusing on the northwestern Indian Ocean, the strong seasonal cycle of the Asian Monsoon has an important impact on the mixed layer of the Arabian Sea as well as on the circulation at the intermediate layer. Between 450-500 m depth the western boundary circulation reverses direction from a strong northward flow in summer (Fig. 5a) to a weaker southward flow in winter (Fig. 5b). For the annual mean this sums up to a northward directed western boundary current, as previously found by Schott and McCreary (2001), with the strongest variability off the Somali and Omani coast (Fig. 5c).

During the summer monsoon enhanced upwelling occurs along the western boundary leading to a region of highest biological productivity. Hence, the core of the ASOMZ is expected at the same place at intermediate depth due to resulting high consumption rates below high productivity rates. However, the core of the ASOMZ is shifted away from that region and is more pronounced in the eastern basin than at the expected area along the western boundary (Fig. 2, 4) (Acharya and Panigrahi, 2016). Suboxic conditions with oxygen concentration of less than 10 µmol/kg are found between 200 and 1000 m depth (Fig. 3) with weak annual variability in the deeper core according to gridded observational data (not shown). The east-west contrast in oxygen concentrations found in the updated WOA 13 data (Fig. 4) confirms the results shown in Resplandy et al. (2012), who used a prior version of the WOA data.

However, oxygen concentrations from monthly mean gridded observations indicate a seasonal variability in the upper level of the ASOMZ showing higher oxygen values in winter and spring inter-monsoon season (Fig. 4a), which are more pronounced in the east. This ventilation goes along with a shallowing of the suboxic layer in April and May (Fig. 4b). This seasonal variability was observed earlier by Sarma (2002) and Banse et al. (2014) showing higher oxygen values in the northern AS at around 300 m depth during the northeast monsoon.

In this study main pathways of the ventilation in the AS are determined as well as their temporal and spatial variability. Based on the mean annual cycle of the oxygen concentration and its layer thickness, the launching locations for the two-dimensional trajectory calculations are defined by placing one point in the eastern basin, where high seasonal variability is observed and the other one in the western part, where the variability of the upper ASOMZ is weaker.

Present results from the trajectory calculations on the isopycnal density layer of 27 kg/m$^3$ reveal a main ventilation pathway of the eastern part of the ASOMZ along the perimeter of the basin (Fig. 7b, d). In late summer and during the autumn inter-monsoon season, RSW spreads out of the Gulf of Aden (Fig. 11d) and flows northward along the coast of Oman, having the same direction as the surface current. In the northwestern part of the basin where the Gulf of Oman merges with the AS, PGW about constantly runs out throughout the year (Fig. 8b, 9b). However, the flow into the eastern basin along the northern boundary peaks during the winter monsoon, turning southward and ventilates the eastern part of the ASOMZ.



A more direct interior pathway, especially from the RS into the eastern basin south of 21°N is negligible confirming previous studies (Lachkar et al., 2016). Tracking the particles of RSW using a water mass analysis Acharya and Panigrahi (2016) showed a maximal percentage of spreading along the coastlines but no propagation of RSW in the interior basin. However, there is weak exchange of water between the eastern and western part of the ASOMZ, which is maximal in May (Fig. 9d).

Two-dimensional histogram maps of particle positions (Fig. 6) also reveal the advection of particles from the southeast into the eastern part of the ASOMZ. The amount of water ventilating on the isopycnal surface from the south is 1.5 times higher than that from the north and strongest during the summer monsoon (Fig. 8e, 9e). During that phase the velocity data (Fig. 5b) reveal a northward flow along the coast of south India that has the opposite direction as the surface current (Hood et al., 2017). A more pronounced ventilation from the south (8°N) in the eastern AS was earlier found by Acharya and Panigrahi

(2016). Even though ICW spreads northward uniformly across the basin at intermediate depth (You and Tomczak, 1993), our results suggest that ventilating particles (Fig. 6) enter the AS predominantly along the eastern boundary (Schott and McCreary, 2001) than along the western boundary, as they do in the thermocline (You and Tomczak, 1993).

The western part is ventilated more equally from around covering particle origins in the northern AS. The RSW that spreads

out of the Gulf of Aden in late summer passes the western basin OMZ on its way northward (Fig. 7c) and thus shows the highest ventilation rate during the autumn inter-monsoon season (Fig. 9c). Ventilation by PGW from the north is strongest during the summer monsoon (Fig. 9a). However, Prasad et al. (2001) stated that PGW spreads further down the Omani coast during winter monsoon with the western boundary undercurrent and more equally to the eastern basin around the northern pathway during the rest of the year. The same spreading patterns of PGW (Fig. 11a, c) can be confirmed also on the

isopycnal surface. Nonetheless the western basin ASOMZ appears to be weakly influenced by that boundary circulation.
A seasonal cycle is also found for RSW (Fig. 11b, d), which spreads southward along the Somali coast during the northeast monsoon, whereas this pathway is not prominent during summer. These results are in agreement with the study of Beal et al. (2000), who tracked RSW spreading by salinity properties.
Calculations of the travel distance show wider spreading of particles for the whole basin during the summer monsoon,

whereby no significant differences in particle probability patterns from the eastern OMZ are noticed after one year in the seasonal spreading comparison (Fig. 10b, d).
Seasonal changes in the western basin are predominantly pronounced in coastal regions than in the centre of the ASOMZ (Fig. 10a, c). This might be due to the reversing boundary current system in the thermocline and intermediate depth (Fig. 5; Schott and McCreary, 2001), which favours higher spreading distances during the southwest monsoon due to stronger

surface currents.
For all releases the calculated travel distances also decrease with time, hypothesizing a recirculation of the fluid parcels that could be eddy driven according to studies of Resplandy et al. (2011), McCreary et al. (2013) and Lachkar et al. (2016), who stated horizontal eddy mixing as a key mechanism for the ventilation of the ASOMZ in their models.



Ventilation pathways from the marginal seas, which are bound to the western basin, are closer to the western part of the OMZ than to the eastern part (Fig. 8) especially for water stemming from the RS. The analysis of a point to point transit time of particles that reach the marginal seas shows that particles travel at least 2 years from the PG into the eastern AS (Fig. 8d) along the perimeter. Particles from the RS travel 5.5 years (Fig. 8d). The calculated time periods of mean ventilation of PGW and RSW are approximately 4 to 5 and >9 years respectively (Tab. 1). These values lie within the range of the residence times of Sarma (2002) and Olson et al. (1993), which vary between 6 to 10 years respectively.

The comparison between different years (Tab. 1) shows a high interannual variability of the ventilation times. Nevertheless, more runs are required to calculate reliable statistics as well as extended time series are needed to confidently predict interannual variabilities.

We conclude that

- The main ventilation pathways for RSW and PGW flow along the perimeter of the basin. A more direct interior pathway for RSW to the eastern basin is not found.

Although ventilation times of particles from the marginal seas into the eastern part of ASOMZ exceed the ones into the western part, prolonged ventilation times are not sufficient to explain the different characteristics in the eastern and the western part of the ASOMZ regarding seasonal variability and vertical extension. During the summer monsoon RSW passes the area of strong primary production off the coast of Oman (Acharya and Panigrahi, 2016). PGW, which enters and ventilates the northern basin predominantly during the winter monsoon, also passes the area of high primary production in the outflow region of the Gulf of Oman (Acharya and Panigrahi, 2016; Lachkar et al., 2018) before it enters the eastern part of the ASOMZ.

- Therefore, the eastward shift of the ASOMZ might be explained by both the resulting longer ventilation time to the eastern part of the OMZ as well as by mixing during the transition of regions of high biological productivity and resulting high consumption rates below at intermediate depth.

Hence, the seasonal variability of oxygen in the eastern basin might be explained by the ventilation of different water masses and different pathways during different monsoon phases. The permanently northward directed flow from the south along the coast of India in the intermediate layer (Fig. 5) seems to be a countercurrent to the changing surface circulation during summer monsoon (Schott and McCreary, 2001; Hood et al., 2017).

The northward flow of Indian Central Water along the Indian coast reveals a seasonal variability, which is strongest during the early summer monsoon and weakest during autumn (Fig. 9f). During the summer monsoon, the Indian Central Water ventilating from the south (Fig. 10e) seems to carry more oxygen than the water coming from the PG and RS during passing regions of high primary production (Acharya and Panigrahi, 2016; Lachkar et al., 2018) before it reaches the eastern ASOMZ. The stronger ventilation of the eastern ASOMZ is mirrored in the decreasing thickness of the layer containing oxygen of less than 10 µmol/kg in May (Fig. 4).




During the winter monsoon, the eastern part of the ASOMZ is highly affected by the transport of water from the north (Fig. 9b), which is low in oxygen due to its long ventilation times as well as its passage of areas of high primary productivity. This is reflected in the thickening of suboxic water in the eastern basin (Fig. 4).

In the western basin the ventilation is strongest during the summer monsoon (Fig. 9a, b) but the oxygen values are higher

during spring (Fig. 4a). Compared to the strong variability of the western boundary current, the oxygen cycle in the western basin is weak. This puzzle could be explained by a loss of oxygen via consumption on the ventilation path leading to a strong summer monsoon transport of oxygen depleted water into the OMZ, which is actually confirmed by Acharya and Panigrahi (2016). Further, mesoscale eddies might play a role in balancing and shaping the ASOMZ as they have an impact on biological production and ventilation rates (McCreary et al., 2013).

•   The seasonal changing intermediate current system along eastern and western boundaries of the Arabian Sea, driven by the monsoon winds, causes the seasonal cycle of the upper ASOMZ.

There are small scale interannual variabilities, whose extent cannot be fully examined here with the comparably short time series of 13 and 6 years, respectively. Certainly, the biannual changing monsoon circulation and eddy mixing have a main impact on the upper volume of the OMZ as well as on the seasonal cycles of the outflow regions, which is strongly coupled

with the seasonal variability of biogeochemical processes, especially in core region of the eastern ASOMZ.

For an error estimation, more runs are required to calculate reliable statistics as well as extended time series are needed to confidently predict interannual variabilities. Nevertheless, the simplified backward trajectory approach seems to be a good method for prediction of the ventilation pathways in the ASOMZ, so it would be interesting to extend such an analysis onto other isopycnal layers, maybe in the deeper water column to see whether there are differences.

**Author contribution**

H. Schmidt, R. Czeschel, and M. Visbeck conceived the study. H. Schmidt handled all the data and performed the simulations. All authors discussed, wrote and modified the manuscript.

**Competing interests**

The authors declare that they have no conflict of interest.

**Data availability**

The 1/12° global HYCOM+NCODA Ocean Reanalysis output is publicly available at http://hycom.org. The WOA13 data are available at https://www.nodc.noaa.gov/OC5/woa13/woa13data.html.



## Acknowledgements

Financial support was received through GEOMAR. This work is a contribution of the Deutsche Forschungsgemeinschaft (DFG) supported project "Sonderforschungsbereich 754: Climate-Biogeochemistry Interactions in the Tropical Ocean" (http://www.sfb754.de). The WOA13 data are available at https://www.nodc.noaa.gov/OC5/woa13/woa13data.html. The

1/12° global HYCOM+NCODA Ocean Reanalysis was funded by the U.S. Navy and the Modelling and Simulation Coordination Office. Computer time was made available by the DoD High Performance Computing Modernization Program. The output is publicly available at http://hycom.org.

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

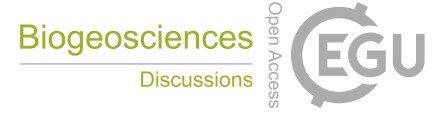



| Run | Launching | Launching date | Calculation Time (years) | Type of calculation | PG percentage | RS percentage | ICW percentage east | west |
|---|---|---|---|---|---|---|---|---|
| 1 | ER | Dec 2012 | 13 | bw | 16.0 | 2.2 | 39.7 | 26.2 |
| 2 | ER | Dec 2012 | 13 | bw | 16.0 | 2.2 | 39.7 | 26.2 |
| 3 | ER | Dec 2012 | 13 | bw | 16.1 | 2.2 | 39.2 | 25.7 |
| 4 | ER | Dec 2012 | 13 | bw | 16.2 | 2.2 | 39.5 | 26.1 |
| 5 | ER | Dec 2012 | 13 | bw | 15.9 | 2.2 | 39.5 | 26.2 |
| 6 | WR | Dec 2012 | 13 | bw | 28.9 | 6.1 | 21.2 | 17.3 |
| 7 | ER | Dec | 13 | bw,clim | 14.6 | 6.4 | 9.0 | 6.6 |
| 8 | WR | Dec | 13 | bw,clim | 7.7 | 11.5 | 13.0 | 12.4 |
| 9 | ER | Dec | 1 | bw,clim | | | | |
| 10 | ER | Sept | 1 | bw,clim | | | | |
| 11 | ER | June | 1 | bw,clim | | | | |
| 12 | ER | Mar | 1 | bw,clim | | | | |
| 13 | WR | Dec | 1 | bw,clim | | | | |
| 14 | WR | Sept | 1 | bw,clim | | | | |
| 15 | WR | June | 1 | bw,clim | | | | |
| 16 | WR | Mar | 1 | bw,clim | | | | |
| 17 | ER | Dec 2010 | 11 | bw | 21.5 | 3.1 | 19.4 | 12.7 |
| 18 | WR | Dec 2010 | 11 | bw | 37.3 | 6.1 | 12.6 | 10.6 |
| 19 | PG | Jan | 1 | fw,clim | | | | |
| 20 | PG | Apr | 1 | fw,clim | | | | |
| 21 | PG | July | 1 | fw,clim | | | | |
| 22 | PG | Oct | 1 | fw,clim | | | | |
| 23 | RS | Jan | 1 | fw,clim | | | | |
| 24 | RS | Apr | 1 | fw,clim | | | | |
| 25 | RS | July | 1 | fw,clim | | | | |
| 26 | SR | Oct | 1 | fw,clim | | | | |



| 27 | ER | Dec 2012 | 13 | bw | 16.5 | 2.3 | 39.2 | 26.4 |
|----|----|----------|----|----|------|-----|------|------|
| 28 | ER | Dec 2012 | 13 | bw | 12.1 | 1.8 | 40.7 | 27.5 |
| 29 | ER | Dec 2012 | 13 | bw | 17.7 | 2.2 | | |
| 30 | ER | Dec 2012 | 8 | bw | 8.4 | 0.8 | 30.3 | 15.4 |
| 31 | ER | Dec 2010 | 8 | bw | 13.3 | 1.3 | 14.0 | 6.9 |
| 32 | ER | Dec 2008 | 8 | bw | 8.2 | 0.8 | 25.7 | 15.7 |
| 33 | WR | Dec 2012 | 8 | bw | 22.7 | 3.9 | 10.2 | 6.7 |
| 34 | WR | Dec 2010 | 8 | bw | 33.8 | 4.6 | 7.0 | 5.5 |
| 35 | WR | Dec 2008 | 8 | bw | 24.3 | 9.4 | 5.4 | 5.5 |
| ER | | 66.64°E | 19.04°N | | | | | |
| WR | | 62.00°E | 19.04°N | | | | | |
| PG | | 59.04°E | 24.00°N | | | | | |
| RS | | 49.04°E | 13.04°N | | | | | |

**Table 1: Trajectory calculation runs as performed for this study and fluid particle percentages with origin in the source regions (PG/RS and southern IO) of the particles to the two launching points (ER/WR). The launch area is defined as a circle with a radius of twice the grid spacing, thus 1/6°, around the launching coordinates (abbreviations defined at the end of the table). The number of released floats is 50000 for runs 1-26 and 10000 for runs 27-35. Except for runs 28 and 29 the diffusion coefficient is 20 m$^2$s$^{-1}$. Abbreviations of calculation type stand for backward- (bw) and forward-in-time (fw) calculation with the whole series of data or an everyday mean over the 13 years computed with a loop (clim).**




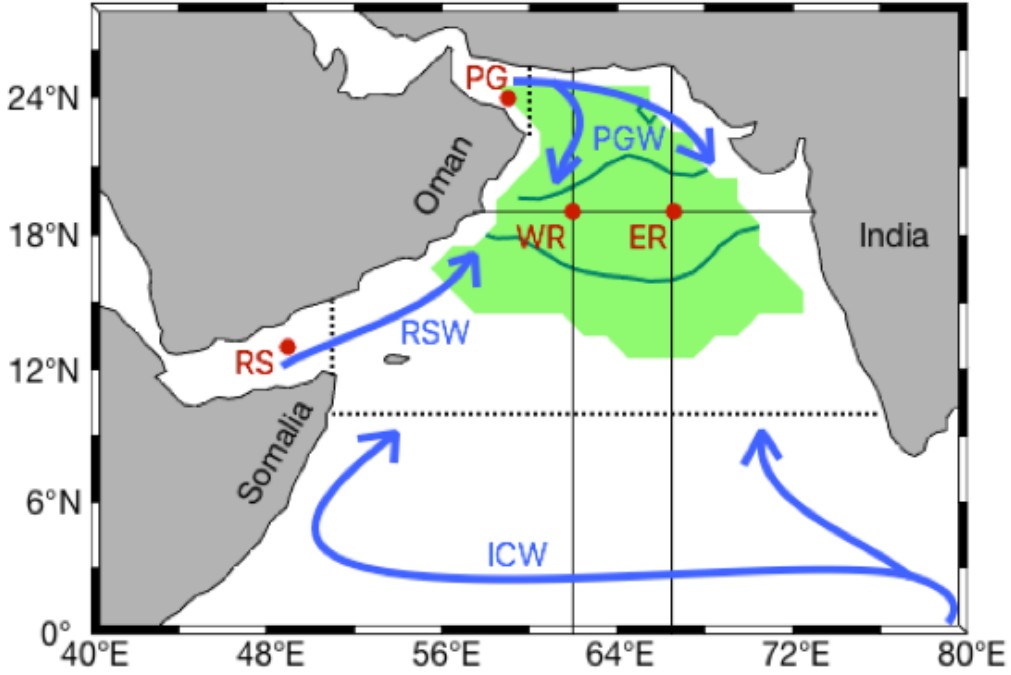

**Figure 1: Northwest Indian Ocean, location of four release points (red) of particles, approximate origin of Indian Central Water (ICW), Red Sea Water (RSW) and Persian Gulf Water (PGW) marked in blue. Black solid lines indicate the sections shown in Fig.**
5 **3. The green patch sketches the location of the Arabian Sea Oxygen Minimum Zone (ASOMZ).**





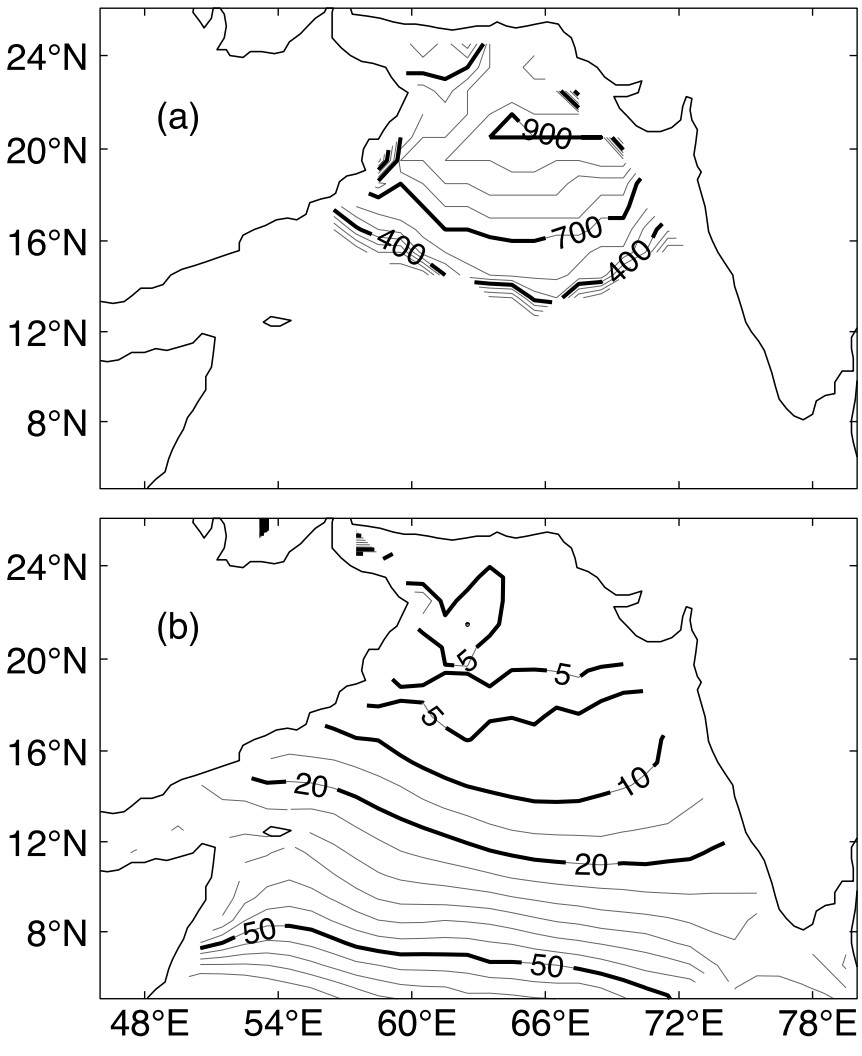

**Figure 2: (a) Thickness (in m) of the layer containing less than 10 µmol kg$^{-1}$ oxygen based on climatological data from WOA 13. (b) Oxygen concentration (in µmol kg$^{-1}$) on the σ = 27 kg m$^{-3}$ isopycnal of WOA 13.**




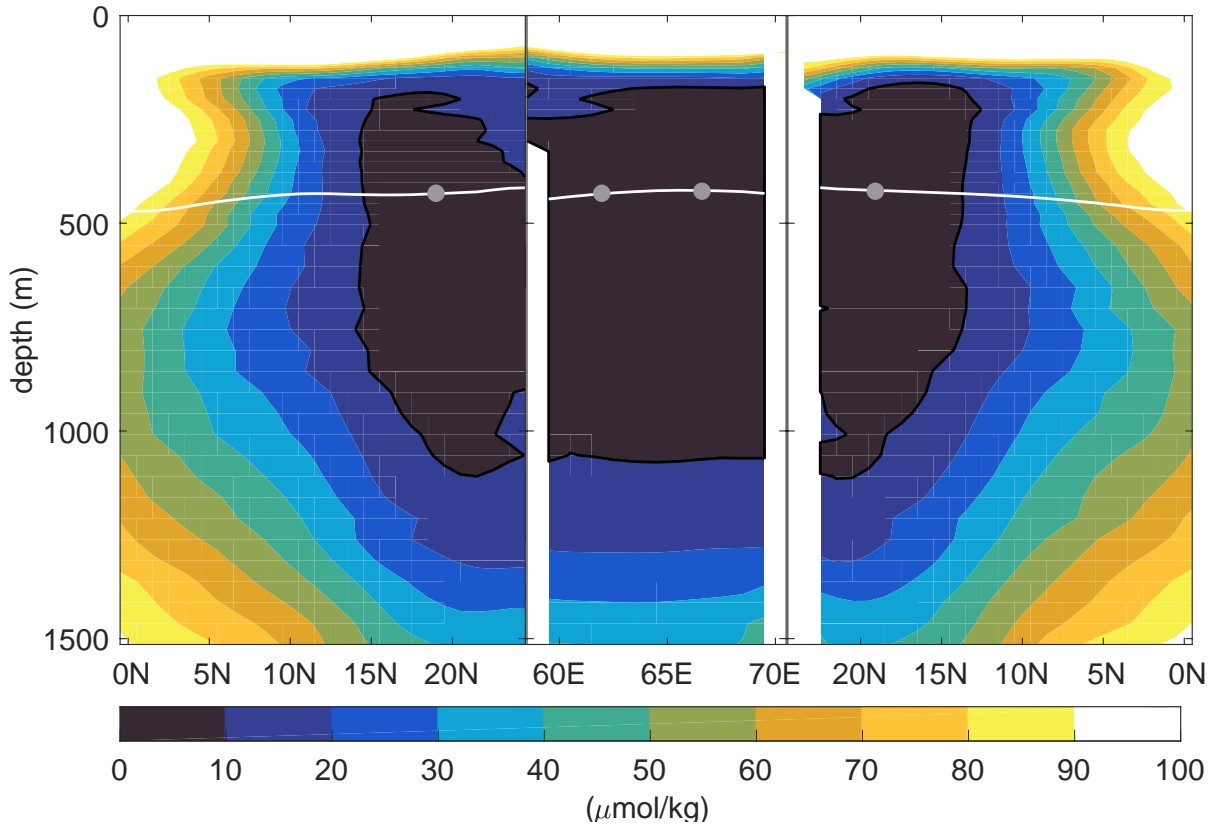

**Figure 3: Annual mean of dissolved oxygen concentration along 62°E (left), 66.5° E (right) and along 19° N (middle) from the WOA 13 climatology (see Figure 1). The white line shows the depth of the σ=27 kg m⁻³ isopycnal. The grey dots mark the release points.**





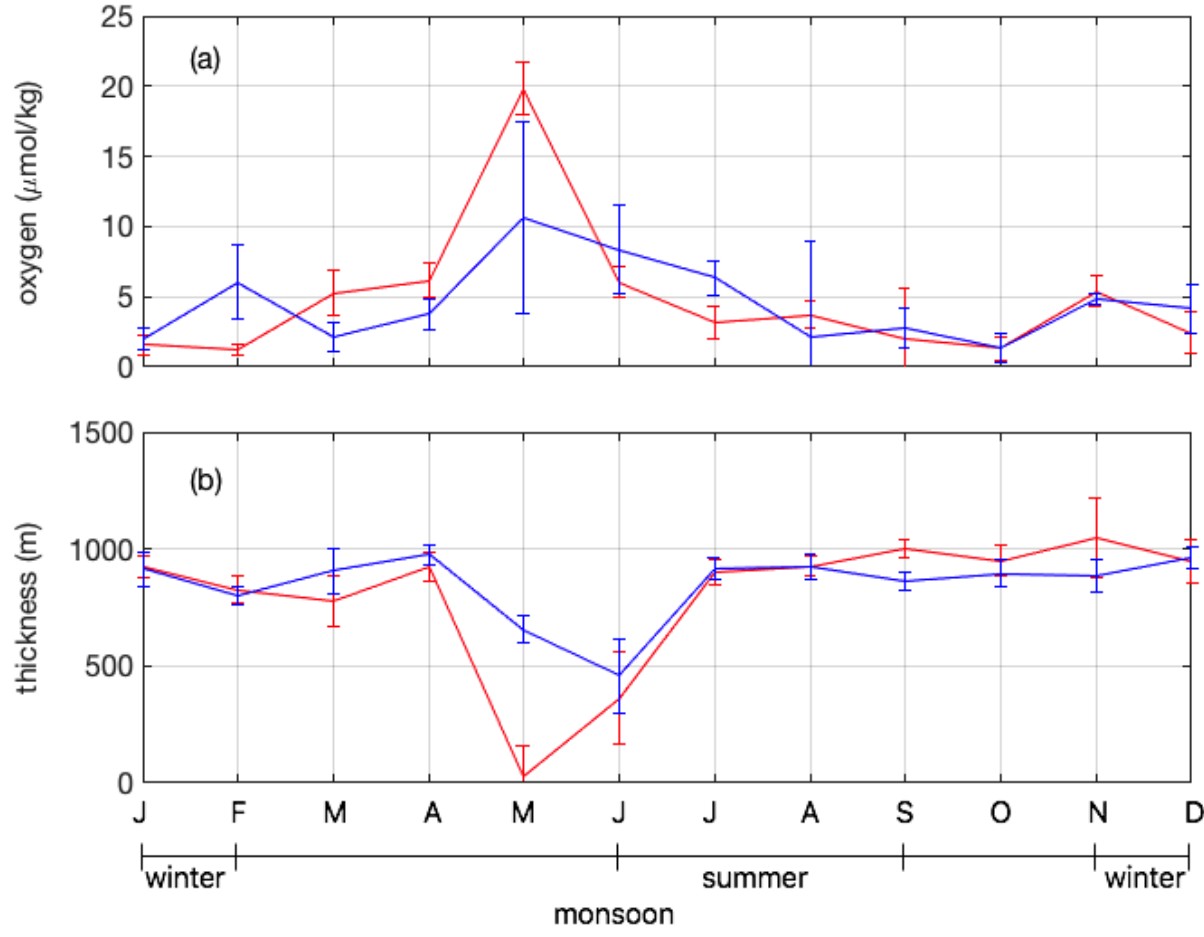

**Figure 4: (a) Mean seasonal cycle of dissolved oxygen concentration at the location of the ER (red) and the WR point (blue) from observations on the isopycnal surface of σ=27 kg m⁻³. (b) Mean seasonal cycle of the thickness of the layer containing oxygen of less than 10 µmol kg⁻¹ based on WOA 13 at the location of the ER (red) and the WR point (blue). The error bars show the spatial standard deviation in an area of 1° around the release point.**





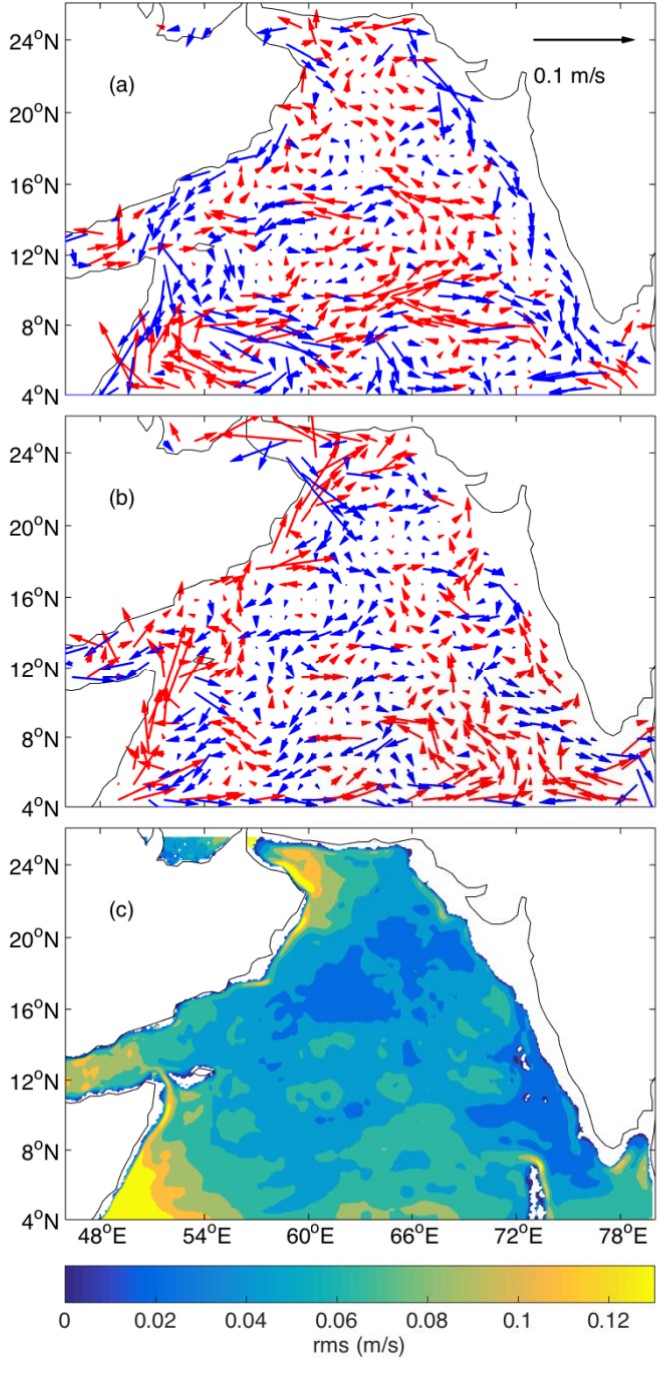

**Figure 5: Mean seasonal velocity for the Arabian Sea based on Hycom data on the isopycnal surface of σ=27 kg m⁻³ for (a) northeast (November-February) and (b) southwest monsoon (June-September) averaged for 2000-2012. The velocity field is spatially filtered (0.6° x 0.6° window) and presented on a grid with the same resolution. Northward (southward) directed currents are shown in red (blue). (c) Root Mean Square (rms) error of the annual mean velocities from 2000-2012.**





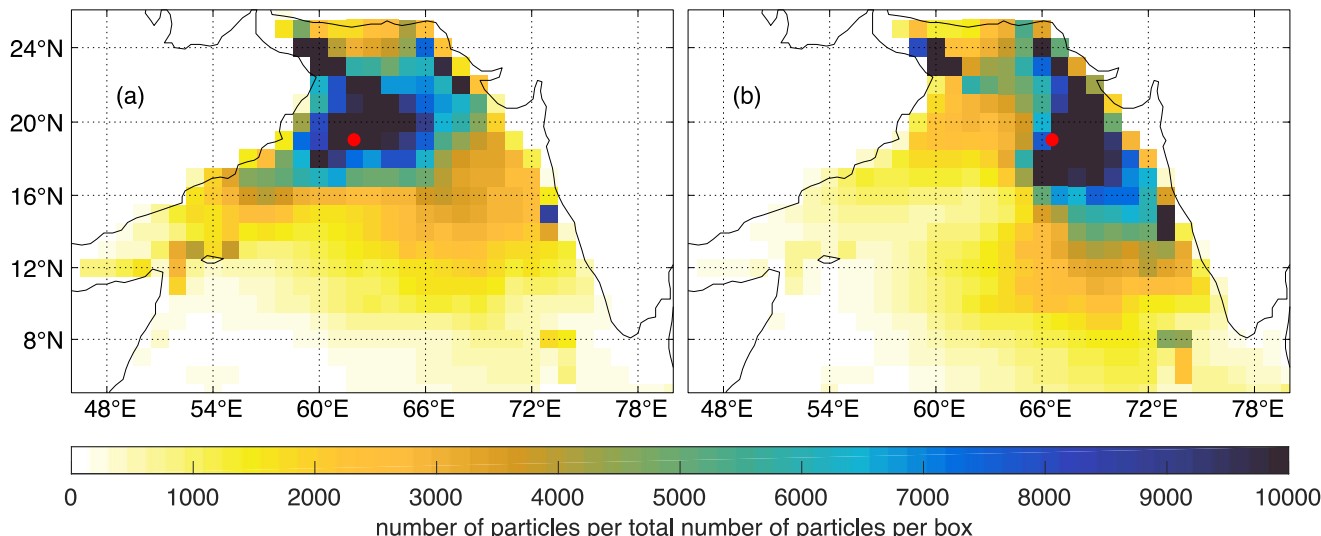

**Figure 6: Two-dimensional maps showing the number of fluid parcels per box over the period of 13 years normalised by the total number of particles per box for backward trajectory calculations from (a) the western (run 6) and (b) the eastern part of the ASOMZ (run 1). Red dots mark the location of the WR and ER points.**





**Figure 7: Particle position maps showing the most pronounced pathways of fluid particles for the backward trajectory analysis, entering the Persian Gulf from (a) WR (run 6) and (b) ER (run 1) and the Red Sea from (c) WR (run 6) and (d) ER (run 1). Eastern (ER) and western release (WR) points in the OMZ are marked in red, as well as the sections to define the source regions.**





**Figure 8:** Point to point transit time of particles across sections (see Section 2.2 for a detailed description): 21° N in dotted dark blue, 17° N in solid purple, 64.3° E in dashed light blue, 10° N in dotted light green (east) and solid green (west), 60° E in solid yellow and 51° E in dotted red as coloured in the map for particles from the RS and PG streaming into the western (c) and eastern (d) OMZ as well as particles from the south streaming into the western (e) and eastern (f) OMZ. The red dot marks the launching position of the backward trajectories (WR, run 33-35; ER, run 30-32).





**Figure 9: Mean seasonal cycle of particle counts across different sections (line colour and type as in Figure 8) for particles from the RS and PG streaming into the western (c) and eastern (d) OMZ as well as particles from the south streaming into the western (e) and eastern (f) OMZ.**







**Figure 10: Two-dimensional histogram maps showing the particle number per box, normalised by the total number of particles per box for equal distribution, for runs 16 (a), 12 (b), 14 (c) and 10 (d). Eastern (ER) and western release (WR) points are marked in red. The calculation time for each run is one year backward, starting in March (a, b) and September (c, d).**





**Figure 11:** Two-dimensional histogram maps showing the particle number per box, normalised by the total number of particles per box for equal distribution, for runs 20 (a), 24 (b), 22 (c) and 26 (d). Release points for the Red Sea (RS) and the Persian Gulf (PG) are marked in red. The calculation time for each run is one year forward, starting in April (a, b) and October (c, d).

