# Peer review of "Ventilation dynamics of the Oxygen Minimum Zone in the Arabian Sea"

_Biogeosciences, 2019_

## Referee Comment (RC1) · Anonymous Referee #1 · 10 Jul 2019

The paper by Schmidt et al. investigates the dynamics of the ventilation of the Arabian Sea and its role in shaping the intensity and variability of the oxygen minimum zone (OMZ) located there. To this end, the study uses Lagrangian trajectories based on a velocity field derived from a reanalysis by the HyCOM model. The authors claim that the eastern part of the OMZ is ventilated mostly from the north by the PGW in winter whereas the western OMZ is ventilated essentially from the southeast during the summer season. The study investigates the ventilation seasonality and timescales as well as the potential role the Arabian marginal seas (the Red Sea and the Persian Gulf) play in this regard.

I) General comment:

The subject of the paper is highly relevant in the general context of understanding the

pathways and timescales of the ventilation of the Arabian Sea and how they impact the OMZ. However, I have several major concerns that prevent me from recommending this manuscript for publication. First, the paper is poorly written. The objectives are not stated clearly and key details of the Lagrangian experiments are missing. Moreover, the explanations given by the authors are sometimes vague or difficult to understand. More importantly, the experimental design does not seem to be appropriate to draw conclusions in a quantitative manner. Below, I develop these points with more specific comments.

II) Specific comments:

1) The study is focused on two sites at 19N (one at 62N and the other at 66.6N). It is not clear what motivates this choice or why are they supposed to represent the dynamics of the whole OMZ?

2) Another point related to the design of the experiment and the robustness of the results is the focus on one particular layer (sigma=27). Why restrict the analysis to this layer if the focus is on the entire OMZ? (especially given the fact that the World's thickest OMZ in the Arabian Sea extends vertically over a wide range of densities from 26 to 27.2)?

3) A related issue is the use of two-dimensional trajectories along one isopycnal surface, failing to take into account upwelling and diapycnal mixing, while these processes may contribute strongly to the ventilation of the OMZ. In particular, we know that the winter convection and water mixing in the north is an important source of ventilation in the northern Arabian Sea. Not being able to take this into account, appears to me to be a major weakness of the study.

4) The details of the particle release experiments are not well described. Are all 50000 particles for each site (ER vs WR) released the same day at the same lat/long point? How is this supposed to capture the spatiotemporal variability around each site?

[Figure]

5) From Table 1, there seem to be large differences in the ventilation sources depending on the duration of integration and the date of particle release (for instance runs 1-5 vs run 7 or run 17). This suggests that the results are not robust with respect to the time of release, and hence they may not necessarily represent the large-scale picture. I suspect the results to be affected by the mesoscale variability around the two sites, which prevents drawing any solid conclusions regarding the ventilation of the Arabian Sea at large.

6) The study focuses on the suboxic core of the OMZ (here defined as O2<10mmol/m3) and uses the World Ocean Atlas (2013) for analysis. Yet, it is known that this dataset strongly overestimates oxygen at low concentrations and hence underestimates the size of the suboxic core of the OMZ and its intensity (see Bianchi et al., 2012 and Banse et al., 2014). Empirical corrections have been proposed to minimize this problem by Bianchi et al (2012) and other studies.

7) The questions of the seasonal maintenance of the OMZ and its eastward shift have been explored by several studies in the past and several drivers have been proposed to explain these observations (e.g., Resplandy et al., 2012, McCreary et al., 2013, Acharya and Panigrahi, 2016). It is not clear what this study adds to what has been proposed before.

8) The model resolution, although in the eddy-resolving range, may not be fine enough to resolve the outflow of the RSW and PGW as these depend on the geometry of narrow straits (especially the Strait of Bab el Mandeb) that requires very high-resolution to be properly represented.

9) Authors claim that the results were insensitive to the choice of the diffusion coefficient. Yet, previous works (see for instance Gnanadesikan et al, 2012, 2013) clearly show that the volume and intensity of OMZs can be very sensitive to the choice of the mixing coefficient. Authors need to explain this.

10) The estimation of ventilation timescales is very confusing. Authors use several

sections very close to the sites of particle release (Fig 8) and focus on short timescales (1year backward and forward experiments, Fig 10 and Fig 11). How can this help to understand the dynamics of the large-scale ventilation of the whole OMZ?

11) Finally, several paragraphs and sections are vague and poorly written. For instance sections 3.2, 3.3 and 4 are not easy to understand.

III) Additional comments:

Fig 7:

The sections are not really located where they should. Not all particles in the Gulf of Aden are originating from the RS nor are all particles in the Gulf of Oman coming from the PG!!!

Fig 8:

What motivated the choice of these sections?

Fig 9:

What is shown in Fig9a and Fig9b? This is not mentioned in the caption.

Figs 10 and Fig 11:

Why restrict the trajectories to 1 year forward/backward?

---

## Referee Comment (RC2) · Anonymous Referee #2 · 18 Aug 2019

**Review of the manuscript "Ventilation dynamics of the oxygen minimum zone in the Arabian Sea" by Schmidt et al., submitted to *Biogeosciences***

This paper claims to investigate the ventilation dynamics of the Arabian Sea oxygen minimum zone (ASOMZ) using a combination of observations and reanalysis velocity fields from HYCOM (Hybrid Coordinate Ocean Model). They perform Lagrangian trajectory analysis based on those velocity fields to understand the pathways of Persian Gulf (PGW), Red Sea (RSW) and Indian Central (ICW) water masses.

The ASOMZ ventilation by the PGW, RSW and ICW have extensively been discussed previously in several studies (the authors themselves cite those papers). It is not clear to me how this paper contributes in terms of new scientific insight and knowledge. I feel that the paper lacks novelty and does not have any new result. I find many inconsistencies throughout the paper and the manuscript is poorly written. Hence I am unable to recommend it for publication in Biogeosciences. Some of the major issues are as follows:

1) I find that the title and abstract of the manuscript are somewhat misleading as the paper is mainly about the Lagrangian pathways of water masses and remains completely decoupled from the oxygen dynamics in the ASOMZ.

2) The paper uses velocity fields from HYCOM, which is a dynamical ocean model. How would you ascertain that the "model dynamics" is actually responsible for the "observed" characteristics of the ASOMZ? It doesn't make sense to me to draw inferences based solely on a physical model to understand the ASOMZ ventilation dynamics. A better approach would be to perform trajectory analysis on the velocity fields from a coupled physical-biogeochemical model that realistically simulates the observed features of the ASOMZ.

3) The velocity fields from HYCOM is the base of their trajectory analysis. Yet the authors provide no validation of these velocity fields. How good are the velocity fields simulated by HYCOM? Are they good enough to capture the ventilation dynamics of the ASOMZ?

4) The entire manuscript is poorly written. Most of the figure captions are not clear and so are the color (Figure 8, for example) legends.

5) The data and method section, which is very difficult for me to follow, does not detail the Lagrangian trajectory computation and experiments. The authors refer to Fischer (2007), which is a Master's thesis and is not readily downloadable as you can see below. How good is this trajectory method compared to the more advanced recent techniques (see Sebille et al.,

Lagrangian ocean analysis: Fundamentals and practices, Ocean Modelling, 121, 2018, p. 49-75, https://doi.org/10.1016/j.ocemod.2017.11.008).

[Figure]

There are jargons used – for e.g. "Spreading distance" - that are not defined in the method section.

6) The justification for the choice of isopycnal layer and the east-west release points are not convincing.

Additional comments:

1. Figure 1: What is the basis for the sketch of ASOMZ? What are those green curves within the green shade?

2. Figure 4: What is the meaning of standard deviation over 1 deg box when the grid size of WOA13 itself is 1 deg? Panel a, comparison of WR (blue) and ER (red) shows higher oxygen in west than in east during June – July (summer). How do you conclude that the ventilation is weaker in west than in the east during summer (Page 7; line 12)? Also, given the error bars in panel a, the ER and WR are not significantly different, I think. Moreover, the prominent seasonal difference in the thickness of suboxic layer, which is 1000 m in the east and 900 m in the west (panel b), is not significantly distinguishable given the error bar range. The text (Page 7, line 8) states that maximum thickness occurs during the winter monsoon with a depth of 1000 m (Fig. 4b) and nearly total oxygen depletion in the core (Fig. 4a). This is not really the case when we look at the figure!  Indeed, fall season (Sep-Nov) shows those characteristics.

3. Figure 8: The discussion in Section 3.3 on this figure is very vague. Wrong reference to Fig. 8e (Page 13, line 7; also for Fig. 9e)

4. Figure 11: for April and October. But it is discussed in the text for summer and winter (page 10, line 32).

5. Page 12: line 17-19: The oxygen concentration is not high during winter, and the statements contradicts seasonality discussed in Section 3.1 and Figure 4a.

6. Inconsistent statements: For example in Page 14, line 2-4: "The analysis of a point to point transit time of particles that reach the marginal seas shows that particles travel at least 2 years from the PG into the eastern AS (Fig. 8d) along the perimeter." In Section 3.3, it is mentioned as 4 years.

7. Loose statements: For example in Page 8, line 21: "The more direct interior pathway is less important". How could you make such statements when there is no quantification of the contribution from pathways.

8. The authors sensitize this study by highlighting the decreasing oxygen trend and an expansion of ASOMZ under climate change (in abstract as well as in the introduction). The two studies that they cite do not specifically illustrate any significant deoxygenation in the entire ASOMZ. Also, some recent studies based on CMIP analysis indicate oxygenation of the southern part of the ASOMZ (Bopp et al., 2013).

---

## Author Comment (AC1) · 30 Nov 2019

The paper by Schmidt et al. investigates the dynamics of the ventilation of the Arabian Sea and its role in shaping the intensity and variability of the oxygen minimum zone (OMZ) located there. To this end, the study uses Lagrangian trajectories based on a velocity field derived from a reanalysis by the HyCOM model. The authors claim that the eastern part of the OMZ is ventilated mostly from the north by the PGW in winter whereas the western OMZ is ventilated essentially from the southeast during the summer season. The study investigates the ventilation seasonality and timescales as well as the potential role the Arabian marginal seas (the Red Sea and the Persian Gulf) play in this regard.

**I) General comment:**

The subject of the paper is highly relevant in the general context of understanding the pathways and timescales of the ventilation of the Arabian Sea and how they impact the OMZ. However, I have several major concerns that prevent me from recommending this manuscript for publication. First, the paper is poorly written. The objectives are not stated clearly and key details of the Lagrangian experiments are missing. Moreover, the explanations given by the authors are sometimes vague or difficult to understand. More importantly, the experimental design does not seem to be appropriate to draw conclusions in a quantitative manner. Below, I develop these points with more specific comments.

Reply to reviewer #1 We would like to thank the reviewer for taking the time and for providing constructive and very specific comments, which helped to improve the manuscript considerably. We have carefully addressed his/her comments. Based on the comments we reworked the passage concerning the design of the Lagrangian experiment in section '2.2 Design of the experiment'. The key details are now explained more precisely and presented more clearly. Also, the data are better presented and results are stated more clearly. In the manuscript we modified several figures (Figs. 3, 6, 7, 8) as well as table 1 and skipped figures 10 and 11. In the supplement we added new figures (Fig. S1, S3, S4, S5) and modified one figure (Fig. S6). The point-by-point responses follow below (written in bold).

II) Specific comments:

1) The study is focused on two sites at 19N (one at 62N and the other at 66.6N). It is not clear what motivates this choice or why are they supposed to represent the dynamics of the whole OMZ?

We shifted the motivation of the choice of the release points to section 2.2 'Design of the experiment': "The contrast in extension and seasonal cycle, not only in oxygen but also in biogeochemical activity (Hood et al., 2009; Resplandy et al., 2012; Brewin et
al., 2012), of the ASOMZ for the eastern and western basin encourages to analyse the ventilation of each half of the basin individually. Therefore, we define two release locations in the eastern (ER) and western (WR) part of the core of the ASOMZ (Fig. 1). The western part is associated with the area of high primary production and the eastern part is associated with the area of lowest oxygen values. Both release areas represent the core of the OMZ and are defined as circles with a radius of twice the grid spacing, thus  $1/6^{\circ}$ , around the launching coordinates, which are  $19.04^{\circ}$  N and  $66.64^{\circ}$  E for the ER and  $19.04^{\circ}$  N and  $62.00^{\circ}$  E for the WR. The Lagrangian particles are spread equally over that area and are all released at the same time (for one run)."

2) Another point related to the design of the experiment and the robustness of the results is the focus on one particular layer (sigma=27). Why restrict the analysis to this layer if the focus is on the entire OMZ? (especially given the fact that the world's thickest OMZ in the Arabian Sea extends vertically over a wide range of densities from 26 to 27.2)?

The careful choice of the isopycnal is based on two reasons that have been described in section '2.2 Trajectory computation', which has been now reworded to section '2.2 Design of the experiment'. In addition we calculated the pathways of the Lagrangian particles on two further isopycnals that are associated with the PGW and RSW. Please see text: "Our experiments are based on the assumption that PGW and RSW are the main local source water masses that are relevant for the ventilation of the ASOMZ and that the oxygen rich waters follow largely their isopycnal layer horizontally. Therefore, advective pathways from Lagrangian particles into the OMZ are calculated on an isopycnal associated with the source regions of the PGW and RSW as well as the OMZ core region. A good representative isopycnal surface of 27 kg/m3 was chosen for most experiments based on two main reasons: The isopycnal lies in the upper core of the ASOMZ (Fig. 3) with low oxygen values of less than 10  $\mu$ mol/kg nearly throughout the entire year (Fig. 4a). Furthermore, this is the density layer with seasonal changes in oxygen concentration (Fig. 4a). The core densities of PGW ( $\sigma = 26.4$  kg/m3) and
RSW ( $\sigma = 27.4$  kg/m3), which appear to be the main source water masses ventilating the ASOMZ, bracket the isopycnal density of  $\sigma = 27$  kg/m3. For the AS, the supply of oxygen was suggested by Banse et al. (2014) to be on the isopycnal surfaces of 27 kg/m3 at depths between 300-500 m depth." ... "Several sensitivity runs were conducted many of them with a reduced number of particles to save computational costs. A comparison between full and reduced number of particles gave very similar results (not shown here). In order to estimate the representativeness of the main Lagrangian pathway analysis on the 27 kg/m3 isopycnal surface two further runs on a shallower and deeper isopycnal were done (for PGW  $\sigma = 26.4$  kg/m3 and RSW  $\sigma = 27.4$  kg/m3). These experiments also used repeated daily velocity data for the calendar year 2006 for costs savings. "The results based the two further runs are presented in the supplement (Figs. S3-S5).

3) A related issue is the use of two-dimensional trajectories along one isopycnal surface, failing to take into account upwelling and diapycnal mixing, while these processes may contribute strongly to the ventilation of the OMZ. In particular, we know that the winter convection and water mixing in the north is an important source of ventilation in the northern Arabian Sea. Not being able to take this into account, appears to me to be a major weakness of the study.

Right, beneath the advection also diffusion as well as consumption affects the oxygen budget. We changed the design of the experiment of the calculation of Lagrangian particle pathways, which are now based on shorter model runs (8 years) in order to estimate the impact of diffusion on the ventilation. This is stated in the manuscript in section '2.2 Design of the experiment': "To estimate the impact of slower diffusion effects on the ventilation of the ASOMZ we compared the typical 8 year long results (Tab.1) to longer 13 year model runs. Again, both experiment gave very similar results pointing towards a secondary role of the slower processes."

Thus, we also changed Figures 6 & 7, which are now based on the 8 year long runs and not on the 13 year long ones. The former results are not influenced by that change:

BGD
"Several sensitivity runs were conducted many of them with a reduced number of particles to save computational costs. A comparison between full and reduced number of particles gave very similar results (not shown here)."

We have also changed the title to "Seasonal variability of the circulation in the Arabian Sea at intermediate depth and its link to the Oxygen Minimum Zone" in order to avoid misunderstanding and to point out that we focus on the advective contribution to the ventilation oxygen minimum zone and its seasonal variability.

4) The details of the particle release experiments are not well described. Are all 50000 particles for each site (ER vs WR) released the same day at the same lat/long point? How is this supposed to capture the spatiotemporal variability around each site?

We addressed the details of the particle release experiments in the new section '2.2 Design of the experiment' and in section '2.4 Trajectory validation and statistics'. Section 2.2: "Both release locations represent the core of the OMZ and are defined as circles with a radius of twice the grid spacing, thus  $1/6^{\circ}$ , around the launching coordinates, which are  $19.04^{\circ}$  N and  $66.64^{\circ}$  E for the ER and  $19.04^{\circ}$  N and  $62.00^{\circ}$  E for the WR. The Lagrangian particles are spread equally over that area and are all released at the same time (for one run). For the forward trajectories two additional release locations in the Gulf of Aden, simulating the spreading of Red Sea Water (RS,  $49.04^{\circ}$  E and  $13.04^{\circ}$  W) and in the Gulf of Oman, simulating the spreading of Persian Gulf Water (PG,  $59.04^{\circ}$  E and  $24.00^{\circ}$  N; Fig. 1) were chosen."

"The Lagrangian particles were advected using the two dimensional velocity fields from HYCOM reanalysis velocity fields followingÂăbasic relations of continuous deformation (see Supplement, Lamb, 1879).ÂăThis approach is consistent with more recent techniques as described in van Sebille et al. (2018). The daily velocity fields were vertically linearly interpolated onto the target isopycnal surface. The number of Lagrangian particles released is 50000 for the runs that were mainly used for statistical purpose (see also Section 2.4) and 10000 for runs 1 to 10 (Tab. 1). The particles were advanced
using an Euler forward-in-time integration scheme using a time step of 1/20 day. Both forward and backward trajectories were calculated and particle positions are stored every 4th day. In addition to the model velocity field a random walk of particles is applied to represent subscale diffusion of 20 m2/s. Near the coast a special case of random walk in the offshore direction is used to prevent trajectories leaving the ocean. The choice of magnitude of random walk is connected to the spatial and temporal grid resolution. A sensitivity experiment with different subscale diffusion coefficients of 10, 20 and 25 m2/s does not reveal significant different results (not shown here). Nevertheless, there are some grid boxes along the coastline and especially near islands, where the particles get trapped. These spuriously high probabilities were not considered for further analyses. Moreover, the velocity fields of HYCOM are obviously divergent, in particular in up and downwelling regions near coasts and islands (e.g. the Maledives, Socotra). Several sensitivity runs were conducted many of them with a reduced number of particles gave very similar results (not shown here)."

Section 2.4: "To test the reliability of the calculated Lagrangian trajectories 5 model runs with identical setup were performed (each with 50000 particles, 13 years duration, starting all at the ER in December 2012). The differences between these runs are discussed in Section 3.5. To detect the interannual variability the runs with the duration of 13 years used for the statistics were compared to a climatological run, which was performed with velocity fields with the mean daily velocity of the 13 years at each grid point and day. Furthermore, the 8 year long runs (runs 1- 6; Tab. 1) were started with a temporal offset of 2 years between the individual runs. For the analysis of seasonality and transit time, we used the mean of these runs to smooth out the interannual variability. Seasonal differences in particle movement around the release locations can be predicted by starting the calculations with a lack of 3 month (January, April, July and October). This was done for forward calculated trajectories from the RS/PG release to predict the spreading op RSW/PGW."
5) From Table 1, there seem to be large differences in the ventilation sources depending on the duration of integration and the date of particle release (for instance runs 1-5 vs run 7 or run 17). This suggests that the results are not robust with respect to the time of release, and hence they may not necessarily represent the large-scale picture. I suspect the results to be affected by the mesoscale variability around the two sites, which prevents drawing any solid conclusions regarding the ventilation of the Arabian Sea at large.

Yes, you are right. There are differences in the ventilation source depending on the year of the particle release. This is due to the interannual variability in the AS and we discussed that point in more detail now. Discussion and Conclusions: "The comparison of travel times and particle amounts between different years (Tab. 1) and also with climatological runs shows high discrepancies and standard deviations which let suspect a strong dependency on interannual variability, that is likely driven by the strength of the monsoon forcing. Another point that underlines the connection between the monsoon forcing and strength and variability of watermass advection into the ASOMZ is the comparison between the 3 isopycnal layers (Tab. 1). With increasing depth the transit times become longer, pointing towards weaker currents and circulation."

6) The study focuses on the suboxic core of the OMZ (here defined as O2

changes not on the absolute values of oxygen. Hence, exact values of oxygen are of minor interest. The objective of the experiment is on the advective pathways of Lagrangian particles relevant for the supply of the ASOMZ and their seasonal variability.

7) The questions of the seasonal maintenance of the OMZ and its eastward shift have been explored by several studies in the past and several drivers have been proposed to explain these observations (e.g., Resplandy et al., 2012, McCreary et al., 2013, Acharya and Panigrahi, 2016). It is not clear what this study adds to what has been proposed before.

This study is based on reanalysis data and the focus of our experiments is on the seasonality of the pathways of Lagrangian particles that have an impact on the upper ASOMZ. We raised these points more clearly in the manuscript by changing the title and also in the discussion: " All in all, the seasonal changing advective pathways into the ASOMZ fit quite well with the weak seasonal oxygen cycle and show clear differences between the eastern and western basin. Thus we conclude that the water mass advection plays a crucial part for the eastward shift of the ASOMZ and might also for the maintenance of low oxygen throughout the year."

8) The model resolution, although in the eddy-resolving range, may not be fine enough to resolve the outflow of the RSW and PGW as these depend on the geometry of narrow straits (especially the Strait of Bab el Mandeb) that requires very high-resolution to be properly represented.

In this study the outflow of the PGW and the RSW is defined by the crossing of the particles of the meridional sections at the entrance of the Gulf of Oman ( $60^{\circ}E$ ) and the Gulf of Aden ( $51^{\circ}E$ ), respectively (see Fig. 7 for the location of the sections), where the model resolution allows to resolve the advection of the Lagrangian particles from the source regions.

9) Authors claim that the results were insensitive to the choice of the diffusion coefficient. Yet, previous works (see for instance Gnanadesikan et al, 2012, 2013) clearly

BGD
show that the volume and intensity of OMZs can be very sensitive to the choice of the mixing coefficient. Authors need to explain this.

For the advective pathways, the choice of the mixing coefficient plays a minor role. (Gnanadesikan,2012). However, we claim that our advective pathways of the particles are insensitive to the subscale diffusion coefficient we add to the trajectory pathways. This must not be related to the mixing coefficient, that is already set in the HYCOM model. We cannot influence that.

10) The estimation of ventilation timescales is very confusing. Authors use several sections very close to the sites of particle release (Fig 8) and focus on short timescales (1year backward and forward experiments, Fig 10 and Fig 11). How can this help to understand the dynamics of the large-scale ventilation of the whole OMZ?

The estimation of ventilation timescales or transit time is now explained in more detail in section '2.3 Trajectory visualization': "The point to point transit time describes the time that each individual Lagrangian particle takes to transit between defined regions (van Sebille et al., 2018). The transit time is analysed along identified main advective pathways into the ASOMZ between distinct sections (see Section 2.2). The transit time is not unique, as different Lagrangian particles might travel between two regions on different ways in different length of time (Phelps et al., 2013). The here discussed transit times are thus defined by the times where 50% of the particles crossed the distinct sections (percentages refer to the total number of Lagrangian particles that have crossed the section after the whole time span of the simulation (8 years). Therefore, no particle is counted twice as only the first crossing time of each particle at each section is detected. Additionally, the seasonal cycle of Lagrangian particles crossing these sections can be determined."

The choice of the sections is also explained in more detail in the new section '2.2 Design of the experiment': "After analysing the pathways of Lagrangian particles within the ASOMZ (Section 3.2) we focused on the seasonal variation of the circulation in the
AS (Section 3.3). To address this question we chose distinct sections along the main advective pathways of the Lagrangian particles (Fig. 1) and calculated the transit times of the particles to get from one region to another for different pathways: two zonal sections are at equal distance south and north of the release locations ( $17^{\circ}$  N,  $21^{\circ}$  N) to investigate the impact of the northeast and southwest monsoon on the advection of the particles, a meridional section separates the eastern and western half of the basin between the release locations at  $64.3^{\circ}$  E to determine the interior circulation, two meridional sections are located at the borders to the Gulf of Oman and Gulf of Aden as the source of the main water masses and another zonal section at  $10^{\circ}$  N serves as a southern boundary of the AS as our research area to get an insight of the inflow from the south and its variation."

We decided to skip Fig. 10 and Fig. 11 as they do not give any additional information in terms of the seasonal variability.

11) Finally, several paragraphs and sections are vague and poorly written. For instance sections 3.2, 3.3 and 4 are not easy to understand.

We carefully rewrote main parts of the sections 2, 3, and 4 and hope that the details of the experiment as well as results and conclusions are now easy to follow and to understand.

III) Additional comments:

Fig 7: The sections are not really located where they should. Not all particles in the Gulf of Aden are originating from the RS nor are all particles in the Gulf of Oman coming from the PG!!!

As stated above we draw inferences about the outflow of the PGW and the RSW by the crossing of Lagrangian particles of the meridional sections at the entrance of the Gulf of Oman ( $60^{\circ}E$ ) and the Gulf of Aden ( $51^{\circ}E$ ), respectively. We are aware, that not all particles from the Gulfs origin in the RS and PG, but all particles from the RS and
PG have to pass through the Gulfs. With the small overflows, it is very difficult to track particles all the way back. Anyway, this does not change the pathways and seasonality in the AS itself.

Fig 8: What motivated the choice of these sections?

The motivation of the section is now stated in section '2.2 Design of the experiment': "After analysing the pathways of Lagrangian particles within the ASOMZ (Section 3.2) we focused on the seasonal variation of the circulation in the AS (Section 3.3). To address this question we chose distinct sections along the main advective pathways of the Lagrangian particles (Fig. 1) and calculated the transit times of the particles to get from one region to another for different pathways: two zonal sections are at equal distance south and north of the release locations (17° N, 21° N) to investigate the impact of the northeast and southwest monsoon on the advection of the particles, a meridional section separates the eastern and western half of the basin between the release locations at 64.3° E to determine the interior circulation, two meridional sections are located at the borders to the Gulf of Oman and Gulf of Aden as the source of the main water masses and another zonal section at 10° N serves as a southern boundary of the AS as our research area to get an insight of the inflow from the south and its variation."

Fig 9: What is shown in Fig9a and Fig9b? This is not mentioned in the caption.

Thanks, we added the missing information to the figure caption (please, see text): "Figure 9: Mean seasonal cycle of particle percentage travelling between distinct sections along their main pathways and their release points in the western basin (left column) and the eastern basin (right column), respectively. Sections are along 21° N shown as dotted dark blue line and 17° N as solid purple line (a, b), 64.3° E as dashed light blue line (b, c, d), 60° E as solid yellow line, 51° E as dotted red line (c, d), 10° N as dotted light green line (east) and solid green line (west) (e, d). For line colour and type please see also Figure 8."
Figs 10 and Fig 11: Why restrict the trajectories to 1 year forward/backward?

We decided to skip Figs. 10 and 11 as they do not give any additional information for the main message of the study. (Anyway, the motivation of these runs has been mentioned in section 2.3 and numbers of these calculations are discussed in the text in section 3.4 of the first version of the script.)

Please also note the supplement to this comment: https://www.biogeosciences-discuss.net/bg-2019-168/bg-2019-168-AC1supplement.pdf

**BGD**
**Fig. 1.** Figure 3: Annual mean of dissolved oxygen concentration along  $62^{\circ}E$  (left),  $66.5^{\circ}E$  (right) and  $19^{\circ}$  N (middle) from the WOA 13 climatology (see Figure 1). Advective pathways from Lagrangian particles a

---

## Author Comment (AC2) · 30 Nov 2019

This paper claims to investigate the ventilation dynamics of the Arabian Sea oxygen minimum zone (ASOMZ) using a combination of observations and reanalysis velocity fields from HYCOM (Hybrid Coordinate Ocean Model). They perform Lagrangian trajectory analysis based on those velocity fields to understand the pathways of Persian Gulf (PGW), Red Sea (RSW) and Indian Central (ICW) water masses. The ASOMZ ventilation by the PGW, RSW and ICW have extensively been discussed previously in several studies (the authors themselves cite those papers). It is not clear to me how this paper contributes in terms of new scientific insight and knowledge. I feel that the paper lacks novelty and does not have any new result. I find many inconsistencies throughout the paper and the manuscript is poorly written. Hence I am unable to recommend it for

publication in Biogeosciences. Some of the major issues are as follows:

Reply to reviewer #2 Thank you for taking your time to carefully read our manuscript and for providing thoughtful and constructive comments. We carefully modified the manuscript considerably as explained below in the reply to the detailed comments. The point-by-point responses follow below (written in bold).

1) I find that the title and abstract of the manuscript are somewhat misleading as the paper is mainly about the Lagrangian pathways of water masses and remains completely decoupled from the oxygen dynamics in the ASOMZ.

We have changed the title to "Seasonal variability of the circulation in the Arabian Sea at intermediate depth and its link to the Oxygen Minimum Zone" in order to avoid misunderstanding and to point out that we focus on the advective contribution to the ventilation of the oxygen minimum zone and its seasonal variability. For modifications of the abstract please see the text:" Abstract. Oxygen minimum zones (OMZs) in the open ocean occur below the surface in regions of weak ventilation and high biological productivity. Very low levels of dissolved oxygen affect marine life and alter biogeochemical cycles. One of the most intense but least understood OMZs in the world ocean is located in the Arabian Sea in a depth range between 300 to 1000 m. An improved understanding of the physical processes that have an impact on the OMZ in the Arabian Sea is necessary for a reliable assessment of its current state and future development. This study uses a combination of observational data as well as reanalysis velocity fields from the ocean model HYCOM (Hybrid Coordinate Ocean Model) to investigate the advective pathways of Lagrangian particles into the Arabian Sea OMZ at intermediate depths. In the eastern basin, the OMZ is strongest during winter monsoon with a core thickness of 1000 m depth and oxygen values of less than 5 mol/kg. The minimum of oxygen concentration might be favoured by a maximum advection of Lagrangian particles that follows the main advective pathway along the perimeter of the basin into the eastern basin of the Arabian Sea during winter monsoon. These Lagrangian particles pass regions of high primary production and respiration contributing

to a transport of low oxygenated water into the eastern part of the OMZ. The maximum of oxygen concentration in the western basin of the Arabian Sea in May coincides with a maximum southward advection of particles along the western boundary during spring intermonsoon supplying the western core of the OMZ with higher oxygenated water. The maximum of oxygen concentration in the eastern basin of the Arabian Sea in May might be associated with the northward inflow of Lagrangian particles across $10°$ N into the Arabian Sea which is highest during spring intermonsoon. The Red Sea outflow of advective particles into the western and eastern basin starts during the summer monsoon associated with the northeastward current during the summer monsoon. Whereas particles from the Persian Gulf advect over the whole year. As the weak seasonal cycle of oxygen concentration in the eastern and western basin can be explained by seasonal changing advective pathways at intermediate depths into the ASOMZ, the simplified backward trajectory approach seems to be a good method for prediction of the seasonality of advective pathways of Lagrangian particles into the ASOMZ."

2) The paper uses velocity fields from HYCOM, which is a dynamical ocean model. How would you ascertain that the "model dynamics" is actually responsible for the "observed" characteristics of the ASOMZ? It doesn't make sense to me to draw inferences based solely on a physical model to understand the ASOMZ ventilation dynamics. A better approach would be to perform trajectory analysis on the velocity fields from a coupled physical-biogeochemical model that realistically simulates the observed features of the ASOMZ.

Based on studies on the equatorial Pacific from global coupled biogeochemical circulation models Dietze and Löptien (2013) point out that poor model performance with respect to oxygen minimum zones is related to a deficient representation of ventilation pathways rather than being associated with a deficient representation of biogeochemical processes (i.e. respiration). This confirms the need for a better understanding of the intermediate circulation in the AS including the pathways of RSW and PGW to understand the associated variability of the ASOMZ. Therefore in this study we focus on the

advective contribution to the ventilation of the oxygen minimum zone. As physical dynamic is not affected by biogeochemical processes, the analysis of advective pathways of virtual Lagrangian particles from a dynamical ocean model seemed obvious to us. Besides biogeochemical models reveal large uncertainties in comparison to observations such as they fail to simulate the eastern shift of the core of the ASOMZ. Nonetheless, a comparison of our results to the ones calculated from a physical-biogeochemical model would be interesting but beyond the scope of this study.

3) The velocity fields from HYCOM is the base of their trajectory analysis. Yet the authors provide no validation of these velocity fields. How good are the velocity fields simulated by HYCOM? Are they good enough to capture the ventilation dynamics of the ASOMZ?

Now we provide a validation of the HYCOM velocity fields: "For a validation of the HYCOM velocity data we compared the near-surface circulation of HYCOM with the climatology of YoMaHa'07, which is based on observational data obtained from Array of Real-time Geostrophic Oceanography (ARGO) floats (Lebedev et al., 2007). This choice is motivated by the lack of observational data at intermediate depths in the Arabian Sea. The comparison of the near-surface circulation during the winter and summer monsoon between HYCOM and ARGO agrees very well which is given in Fig. S1 in the Supplement. The complex circulation pattern at the near-surface which is strongly affected by the seasonal Asian monsoon is well described by the HYCOM data reflecting all (reversing) currents that are relevant for the AS."

4) The entire manuscript is poorly written. Most of the figure captions are not clear and so are the color (Figure 8, for example) legends.

We carefully rewrote main parts of the sections 2, 3, and 4, modified the figure captions and changed figure 8 to be more comprehensible.

5) The data and method section, which is very difficult for me to follow, does not detail the Lagrangian trajectory computation and experiments. The authors refer to Fischer

(2007), which is a Master's thesis and is not readily downloadable as you can see below. How good is this trajectory method compared to the more advanced recent techniques (see Sebille et al., Lagrangian ocean analysis: Fundamentals and practices, Ocean Modelling, 121, 2018, p. 49-75, https://doi.org/10.1016/j.ocemod.2017.11.008). There are jargons used – for e.g. "Spreading distance" - that are not defined in the method section.

We have inserted a new section '2.2 Design of the experiment' which describes the key details of the calculation of Lagrangian particles more precisely. "2.2 Design of the experiment In the following we would like to present motivation and key details of the conceptual design of the experiments to investigate the main advective pathways within the OMZ of the AS and its seasonal variability. To estimate the advective contribution to the OMZ ventilation we analysed the pathways of Lagrangian particles released into a two dimensional (along isopycnal) model based velocity field. This approach is time efficient, focusses on the isopycnal advection but ignores for example the effects of upwelling or diapycnal mixing. However, this method allows both to estimate contribution from different source regions to the OMZ by performing backward trajectories and to draw inferences of the basin wide spread of oxygen at intermediate levels. Our experiments are based on the assumption that PGW and RSW are the main local source water masses that are relevant for the ventilation of the ASOMZ and that the oxygen rich waters follow largely their isopycnal layer horizontally. Therefore, advective pathways from Lagrangian particles into the OMZ are calculated on an isopycnal associated with the source regions of the PGW and RSW as well as the OMZ core region. A good representative isopycnal surface of 27 kg/m3 was chosen for most experiments based on two main reasons: The isopycnal lies in the upper core of the ASOMZ (Fig. 3) with low oxygen values of less than 10 $\mu$mol/kg nearly throughout the entire year (Fig. 4a). Furthermore, this is the density layer with seasonal changes in oxygen concentration (Fig. 4a). The core densities of PGW ($\sigma$ = 26.4 kg/m3) and RSW ($\sigma$ = 27.4 kg/m3), which appear to be the main source water masses ventilating the ASOMZ, bracket the isopycnal density of $\sigma$ = 27 kg/m3. For the AS, the supply of oxygen was suggested

by Banse et al. (2014) to be on the isopycnal surfaces of 27 kg/m3 at depths between 300-500 m depth.

The contrast in extension and seasonal cycle, not only in oxygen but also in biogeo-chemical activity (Hood et al., 2009; Resplandy et al., 2012; Brewin et al., 2012), of the ASOMZ for the eastern and western basin encourages to analyse the ventilation of each half of the basin individually. Therefore, we define two release locations in the eastern (ER) and western (WR) part of the core of the ASOMZ (Fig. 1). The western part is associated with the area of high primary production and the eastern part is as-sociated with the area of lowest oxygen values. Both release locations represent the core of the OMZ and are defined as circles with a radius of twice the grid spacing, thus $1/6°$, around the launching coordinates, which are $19.04°$ N and $66.64°$ E for the ER and $19.04°$ N and $62.00°$ E for the WR. The Lagrangian particles are spread equally over that area and are all released at the same time (for one run). For the forward tra-jectories two additional release locations in the Gulf of Aden, simulating the spreading of Red Sea Water (RS, $49.04°$ E and $13.04°$ W) and in the Gulf of Oman, simulating the spreading of Persian Gulf Water (PG, $59.04°$ E and $24.00°$ N; Fig. 1) were chosen.

After analysing the pathways of Lagrangian particles within the ASOMZ (Section 3.2) we focused on the seasonal variation of the circulation in the AS (Section 3.3). To address this question we chose distinct sections along the main advective pathways of the Lagrangian particles (Fig. 1) and calculated the transit times of the particles to get from one region to another for different pathways: two zonal sections are at equal distance south and north of the release locations ($17°$ N, $21°$ N) to investigate the impact of the northeast and southwest monsoon on the advection of the particles, a meridional section separates the eastern and western half of the basin between the release locations at $64.3°$ E to determine the interior circulation, two meridional sections are located at the borders to the Gulf of Oman and Gulf of Aden as the source of the main water masses and another zonal section at $10°$ N serves as a southern boundary of the AS as our research area to get an insight of the inflow from the south

and its variation.

The Lagrangian particles were advected using the two dimensional velocity fields from HYCOM reanalysis velocity fields followingÂăbasic relations of continuous deformation (see Supplement, Lamb, 1879).ÂăThis approach is consistent with more recent techniques as described in van Sebille et al. (2018). The daily velocity fields were vertically linearly interpolated onto the target isopycnal surface. The number of Lagrangian particles released is 50000 for the runs that were mainly used for statistical purpose (see also Section 2.4) and 10000 for runs 1 to 10 (Tab. 1). The particles were advanced using an Euler forward-in-time integration scheme using a time step of 1/20 day. Both forward and backward trajectories were calculated and particle positions are stored every 4th day. In addition to the model velocity field a random walk of particles is applied to represent subscale diffusion of 20 m2/s. Near the coast a special case of random walk in the offshore direction is used to prevent trajectories leaving the ocean. The choice of magnitude of random walk is connected to the spatial and temporal grid resolution. A sensitivity experiment with different subscale diffusion coefficients of 10, 20 and 25 m2/s does not reveal significant different results (not shown here). Nevertheless, there are some grid boxes along the coastline and especially near islands, where the particles get trapped. These spuriously high probabilities were not considered for further analyses. Moreover, the velocity fields of HYCOM are obviously divergent, in particular in up and downwelling regions near coasts and islands (e.g. the Maledives, Socotra). Several sensitivity runs were conducted many of them with a reduced number of particles to save computational costs. A comparison between full and reduced number of particles gave very similar results (not shown here). In order to estimate the representativeness of the main Lagrangian pathway analysis on the 27 kg/m3 isopycnal surface two further runs on a shallower and deeper isopycnal were done (for PGW $\sigma$ = 26.4 kg/m3 and RSW $\sigma$ = 27.4 kg/m3). These experiments also used repeated daily velocity data for the calendar year 2006 for costs savings. To estimate the impact of slower diffusion effects on the ventilation of the ASOMZ we compared the typical 8 year long results (Tab.1) to longer 13 year model runs. Again, both experiment gave

very similar results pointing towards a secondary role of the slower processes."

Methods and equations used for the calculation of two dimensional advective pathways of Lagrangian particles are now described in the supplement: Please see supplement (Lamb, S. H.: Hydrodynamics. University Press, Cambridge, 1879.)

We also refer to advanced recent techniques (van Sebille et al., 2018) and use more common jargon. Please see section '2.3 Trajectory visualization': "To analyse the Lagrangian data, the AS is divided into a grid of 1° x 1° resolution. For each time step the number of particles residing in a certain grid box can be counted leading to a map that shows the particle concentration over the analysed time in a certain grid box or at individual time steps (Gary et al., 2011). For a better comparison, the probability for each bin has been obtained. Summing up all particle counts in a certain grid box over the whole time and dividing it by the total number of particle counts for all grid boxes leads to probability maps that sum up to 100% for the whole experimental area and time (van Sebille et al., 2018). With a subsample of the trajectories that reach the source regions, these maps can highlight the most likely advective pathways of Larangian particles (Gary et al., 2011). Additionally, it is possible to analyse the spreading of the particles by looking at single time steps. The point to point transit time describes the time that each individual Lagrangian particle takes to transit between defined regions (van Sebille et al., 2018). The transit time is analysed along identified main advective pathways into the ASOMZ between distinct sections (see Section 2.2). The transit time is not unique, as different Lagrangian particles might travel between two regions on different ways in different length of time (Phelps et al., 2013). The here discussed transit times are thus defined by the times where 50% of the particles crossed the distinct sections (percentages refer to the total number of Lagrangian particles that have crossed the section after the whole time span of the simulation (8 years). Therefore, no particle is counted twice as only the first crossing time of each particle at each section is detected. Additionally, the seasonal cycle of Lagrangian particles crossing these sections can be determined."

6) The justification for the choice of isopycnal layer and the east-west release points are not convincing.

The new chapter 2.2 Design of the experiments clarifies all questions on the motivation for the choice of certain isopycnal layers and the location of the release of the particles. Please see text: "In the following we would like to present motivation and key details of the conceptual design of the experiments to investigate the main advective pathways within the OMZ of the AS and its seasonal variability. To estimate the advective contribution to the OMZ ventilation we analysed the pathways of Lagrangian particles released into a two dimensional (along isopycnal) model based velocity field. This approach is time efficient, focusses on the isopycnal advection but ignores for example the effects of upwelling or diapycnal mixing. However, this method allows both to estimate contribution from different source regions to the OMZ by performing backward trajectories and to draw inferences of the basin wide spread of oxygen at intermediate levels. Our experiments are based on the assumption that PGW and RSW are the main local source water masses that are relevant for the ventilation of the ASOMZ and that the oxygen rich waters follow largely their isopycnal layer horizontally. Therefore, advective pathways from Lagrangian particles into the OMZ are calculated on an isopycnal associated with the source regions of the PGW and RSW as well as the OMZ core region. A good representative isopycnal surface of 27 kg/m3 was chosen for most experiments based on two main reasons: The isopycnal lies in the upper core of the ASOMZ (Fig. 3) with low oxygen values of less than 10 $\mu$mol/kg nearly throughout the entire year (Fig. 4a). Furthermore, this is the density layer with seasonal changes in oxygen concentration (Fig. 4a). The core densities of PGW ($\sigma$ = 26.4 kg/m3) and RSW ($\sigma$ = 27.4 kg/m3), which appear to be the main source water masses ventilating the ASOMZ, bracket the isopycnal density of $\sigma$ = 27 kg/m3. For the AS, the supply of oxygen was suggested by Banse et al. (2014) to be on the isopycnal surfaces of 27 kg/m3 at depths between 300-500 m depth.

The contrast in extension and seasonal cycle, not only in oxygen but also in biogeochemical activity (Hood et al., 2009; Resplandy et al., 2012; Brewin et al., 2012), of the ASOMZ for the eastern and western basin encourages to analyse the ventilation of each half of the basin individually. Therefore, we define two release locations in the eastern (ER) and western (WR) part of the core of the ASOMZ (Fig. 1). The western part is associated with the area of high primary production and the eastern part is associated with the area of lowest oxygen values. Both release locations represent the core of the OMZ and are defined as circles with a radius of twice the grid spacing, thus $1/6°$, around the launching coordinates, which are $19.04°$ N and $66.64°$ E for the ER and $19.04°$ N and $62.00°$ E for the WR. The Lagrangian particles are spread equally over that area and are all released at the same time (for one run). For the forward trajectories two additional release locations in the Gulf of Aden, simulating the spreading of Red Sea Water (RS, $49.04°$ E and $13.04°$ W) and in the Gulf of Oman, simulating the spreading of Persian Gulf Water (PG, $59.04°$ E and $24.00°$ N; Fig. 1) were chosen."

Additional comments: 1. Figure 1: What is the basis for the sketch of ASOMZ? What are those green curves within the green shade?

We removed the green curves from the figure (which have been 5 mol/kg contour lines) to have a plainer sketch and a precise description is now given in the figure caption: Figure 1: The green patch sketches the location of the Arabian Sea Oxygen Minimum Zone (ASOMZ) defined by oxygen of less than 10 mol/kg. Schematic pathways of the three major intermediate source water masses in the northwest Indian Ocean are marked in blue: Indian Central Water (ICW), Red Sea Water (RSW), and Persian Gulf Water (PGW). Location of four particle release points (western basin (WR), eastern basin (ER), Persian Gulf (PG), Red Sea (RS)) are shown as red dots. Black solid lines indicate the sections shown in Fig. 3. Sections, that need to be crossed by the Lagrangian particles to define the source regions for the ICW, RSW, and PGW are marked as black dashed lines.

2. Figure 4: What is the meaning of standard deviation over 1 deg box when the grid size of WOA13 itself is 1 deg?

This is clarified in the figure caption and the text: "Figure 4: ... The error bars show the spatial standard deviation in an area of 2° x 2° centered around the release point."

Panel a, comparison of WR (blue) and ER (red) shows higher oxygen in west than in east during June – July (summer). How do you conclude that the ventilation is weaker in west than in the east during summer (Page 7; line 12)? Also, given the error bars in panel a, the ER and WR are not significantly different, I think. Moreover, the prominent seasonal difference in the thickness of suboxic layer, which is 1000 m in the east and 900 m in the west (panel b), is not significantly distinguishable given the error bar range. The text (Page 7, line 8) states that maximum thickness occurs during the winter monsoon with a depth of 1000 m (Fig. 4b) and nearly total oxygen depletion in the core (Fig. 4a). This is not really the case when we look at the figure! Indeed, fall season (Sep-Nov) shows those characteristics.

We rewrote several sentences concerning Fig. 4. Please see text: "A similar seasonal cycle is prominent in the western AS with a maximum thickness of the suboxic layer of 900 m (Fig. 4b) but however, a weaker ventilation during the spring intermonsoon compared to the east. The layer containing oxygen of less than 10 $\mu$mol/kg remains thicker than 500 m. Based on an area of 2°x 2° in total centered around the release location the spatial standard deviations were calculated (Fig. 4). They show that the seasonal cycle of the OMZ represents a large area and not only the release location. This holds especially for the eastern basin. "

"The maximum thickness arises during fall intermonsoon and at the beginning of winter monsoon with a depth of 1000 m (Fig. 4b) and nearly total oxygen depletion in the core (Fig. 4a). Oxygen concentration increases within spring intermonsoon and the suboxic layer in the eastern AS nearly vanishes in May (Fig. 4b)."

The error bars mark spatial deviations. You are right that they overlap during several month, especially in the winter monsoon. Pleas see the rewrote sentence and figure caption above for Fig. 4.

3. Figure 8: The discussion in Section 3.3 on this figure is very vague. Wrong reference to Fig. 8e (Page 13, line 7; also for Fig. 9e)

Thank you for mentioning that. The discussion gets to the point now. The reference has been corrected. "After analysing the main pathways, in the following we focus on the point to point transit time of the advective Lagrangian particles helping to further understand the circulation at intermediate depth. Therefore, the point to point transit time of the particles across selected sections along their distinct pathways (see section 2.2 for location of the sections) is analysed. As transit time is individual for each particle Fig. 8 shows the cumulative transit time of all particles crossing that section on the isopycnal surface of 27 kg/m3. Additionally, the times where 50% of the particles crossed the distinct sections (percentages refer to the total number of Lagrangian particles that have crossed the section after the whole time span of the simulation (8 years)) are listed in Tab. 1. The western part of the ASOMZ is ventilated preferably from particles coming from the northern basin. Within the first year about 60% of the released particles travel the pathway northward along the western boundary between the 21° N section and the WR (Fig. 8a). The number of particles travelling northward over the section at 17°N is much smaller. Barely 5% of all particles cross that section during the whole calculation time (Fig. 8a). In contrast, in the eastern basin the numbers of particles ventilating the eastern part of the OMZ over the northern and the southern section are about the same with rates of 52% (17°N) and 62% (21°N) of the released particles crossing over the 8 years of calculation time (Fig. 8b). Compared to the western basin (Fig. 8a) the slope of the cumulative particle transit time curve is flatter (Fig. 8b). Thus, the point to point transit times of the individual particles are spread over a wider and longer time range for the ER. 28% of the released particles are travelling around the perimeter of the basin (Fig. 8b), which is roughly 10% more than particles taking the interior pathway between ER and WR (Fig. 8d). However, the exchange between WR and ER in the other direction is more pronounced (Fig. 8c). The point to point transit times for all these sections are less than six month for the fastest particles and the slope of the cumulative transit time is weak, especially for particles released in

the eastern ASOMZ (Fig. 8b, d), pointing towards large differences in transit times for individual particles. Transit times from the PG as well as from the RS are shorter to the western basin (Fig. 8c) than to the eastern basin (Fig. 8d, Tab.1). The mean transit time of 50% of the particles on the isopycnas surface of 27 kg/m3 between the WR and the section in the Gulf of Oman is 2 years (for values of the single runs see Tab. 1). The equivalent mean transit time for the ER is 4.2 years. The mean point to point transit times from the release locations to the RS section for 50 % of the particles are 6.4 and 5.2 years for the ER and WR, respectively. Anyway, the slope of the curves is somehow constant over the whole calculation period, especially for the transit times to the RS (Fig. 8c, d). The AS is also ventilated from the south across 10° N (Fig. 8e, f). For both release points, WR and ER, the ventilation is stronger across the eastern half of the basin (Fig. 8e, f). Here again, the slope of the curves is somehow constant over the whole calculation period. Mean transit times between the release locations and the south eastern section of the basin for 50 % of the particles are 4.8 and 6.0 years for the ER and WR and 5.4 and 5.6 years between the south western section and the ER and WR, respectively. The point to point transit times at the isopycnal surface of 26.4 kg/m3 are nearly entirely quicker compared to the ones that were discussed above for the 27 kg/m3 isopycnal surface (Tab. 1). This tendency extends further down, as the deepest considered Isopycnal surface of 27.4 kg/m3 has the slowest transit times."

4. Figure 11: for April and October. But it is discussed in the text for summer and winter (page 10, line 32).

Thank you for pointing that out. We decided to skip Figs. 10 and 11 as they do not give any additional information for the main message of the study.

5. Page 12: line 17-19: The oxygen concentration is not high during winter, and the statements contradicts seasonality discussed in Section 3.1 and Figure 4a.

We are sorry, we clarified the sentences. Please see text. "However, oxygen concentrations from monthly mean gridded observations indicate a seasonal variability in

the upper level of the ASOMZ showing higher oxygen values in spring intermonsoon (Fig. 4a), which are more pronounced in the east and slightly higher oxygen at the end of the winter monsoon only for the western basin. This ventilation goes along with a shallowing of the suboxic layer in May and June (Fig. 4b)."

6. Inconsistent statements: For example in Page 14, line 2-4: "The analysis of a point to point transit time of particles that reach the marginal seas shows that particles travel at least 2 years from the PG into the eastern AS (Fig. 8d) along the perimeter." In Section 3.3, it is mentioned as 4 years.

With the change of the point to point transit time definition with percentage quantiles and the modification of Fig. 8, we rewrote the paragraph. "Advective pathways from the marginal seas, which are bound to the western basin, are shorter to the western part of the OMZ than to the eastern part, especially for water stemming out of the Gulf of Aden. The analysis of a point to point transit time of particles that reach the marginal seas shows that the mean transit time for 50% of the particles that travel between the PG and the ER is 4.2 years but just 2 years for the WR. Particles from the RS have a mean point to point travel time of 6.4 and 5.2 years to the ER and WR, respectively. However, prolonged transit times alone are not sufficient to explain the different characteristics in the eastern and the western part of the ASOMZ especially when considering the strong seasonal variability of the advective pathways."

7. Loose statements: For example in Page 8, line 21: "The more direct interior pathway is less important". How could you make such statements when there is no quantification of the contribution from pathways.

We rewrote the sentence: "Most particles spread further north along the coastline of Pakistan and India to enter the eastern basin, whereas the more direct interior pathway is less frequent as stated by the percentage of particle (Fig. 7d)."

8. The authors sensitize this study by highlighting the decreasing oxygen trend and an expansion of ASOMZ under climate change (in abstract as well as in the introduction).

The two studies that they cite do not specifically illustrate any significant deoxygenation in the entire ASOMZ. Also, some recent studies based on CMIP analysis indicate oxygenation of the southern part of the ASOMZ (Bopp et al., 2013).

The two studies are cited in a context that generally reveals implications of a deoxygenated ocean. Additionally, we added references on observational studies that specifically describe deoxygenation and follow-ups in the northern Arabian Sea over the last decades (Ito et al. 2017; Queste et al., 2018; Piontkovski and Al-Oufi, 2015). "Observations reveal an intensification of the northern part of the ASOMZ over the period of the last three (Queste et al., 2018) to five decades (Ito et al., 2017) and a shoaling of the hypoxic boundary in the Se of Oman (Piontkovski and Al-Oufi, 2015)." We added the following studies to the reference list: Ito, T., Minobe, S., Long, M. C., and Deutsch, C.: Upper ocean O2 trends 1958-2015, Geophysical Research Letters, 44,4214-4223, https://doi.org/10.1002/2017GL073613, 2017. Piontkovski, S. A. and Al-Oufi, H. S.: The Oman shelf hypoxia and the warming Arabian Sea: International Journal of Environmental Studies, http://dx.doi.org/10.1080/00207233.2015.1012361, 2015. Queste, B. Y., Vic, C., Heywood, K. J., and Piontkovski, S. A.: Physical controls on oxygen distribution and denitrification potential in the north west Arabian Sea. Geophysical Research Letters, 45,4143-4152. https://doi.org/10.1029/2017GL076666, 2018.

Please also note the supplement to this comment:
https://www.biogeosciences-discuss.net/bg-2019-168/bg-2019-168-AC2-supplement.pdf
* * *
[Figure]

**Fig. 1.** Figure 1: The green patch sketches the location of the Arabian Sea Oxygen Minimum Zone (ASOMZ) defined by oxygen of less than 10 mol/kg. Schematic pathways of the three major intermediate source water

**Fig. 2.** Figure 3: Annual mean of dissolved oxygen concentration along 62°E (left), 66.5° E (right) and 19° N (middle) from the WOA 13 climatology (see Figure 1). Advective pathways from Lagrangian particles a

[Figure]

**Fig. 3.** Figure 4: (a) Mean seasonal cycle of dissolved oxygen concentration on the isopycnal surface of $\sigma$=27 kg/m3 at the location of the ER (red) and the WR point (blue) from observations. (b) Mean seasonal

[Figure]

**Fig. 4.** Figure 8: Cumulative point to point transit times of Lagrangian particles calculated between distinct sections (see maps) along their main pathways and their release points in the western basin (left

[Figure]